# Results from a Multi-Laboratory Ocean Metaproteomic Intercomparison:

# Effects of LC-MS Acquisition and Data Analysis Procedures

*Participants of the Ocean Metaproteome Intercomparison Consortium:*

Mak A. Saito[*1], Jaclyn K. Saunders[1+], Matthew R. McIlvin[1], Erin M. Bertrand[2], John A. Breier[3], Margaret Mars Brisbin[1], Sophie M. Colston[4], Jaimee R. Compton[4], Tim J. Griffin[5], W. Judson Hervey[4], Robert L. Hettich[6], Pratik D. Jagtap[5], Michael Janech[7], Rod Johnson[8], Rick Keil[9], Hugo Kleikamp[10], Dagmar Leary[4], Lennart Martens[17,18], J. Scott P. McCain[2,11], Eli Moore[12], Subina Mehta[5], Dawn M. Moran[1], Jacqui Neibauer[7], Benjamin A. Neely[13], Michael V. Jakuba[1], Jim Johnson[5], Megan Duffy[7], Gerhard J. Herndl[14], Richard Giannone[6], Ryan Mueller[15], Brook L. Nunn[9], Martin Pabst[9], Samantha Peters[6], Andrew Rajczewski[5], Elden Rowland[2], Brian Searle[16], Tim Van Den Bossche[17,18], Gary J. Vora[4], Jacob R. Waldbauer[19], Haiyan Zheng[20], Zihao Zhao[14]

[1]Woods Hole Oceanographic Institution, Woods Hole, MA, USA
[2]Department of Biology, Dalhousie University, Halifax, NS, Canada
[3]The University of Texas Rio Grande Valley, Edinburg, TX
[4]Center for Bio/Molecular Science & Engineering, Naval Research Laboratory, Washington, DC, USA
[5]University of Minnesota at Minneapolis, Minneapolis, Minnesota, USA
[6]Oak Ridge National Laboratory, Oak Ridge, Tennessee, USA
[7]College of Charleston, Charleston, South Carolina, USA
[8]Bermuda Institute of Ocean Sciences, Bermuda
[9]University of Washington, Seattle, Washington, USA
[10]Department of Biotechnology, Delft University of Technology, Netherlands
[11]Department of Biology, Massachusetts Institute of Technology, Cambridge, MA, USA
[12] United States Geological Survey, USA
[13]National Institute of Standards and Technology, Charleston, South Carolina, USA
[14]University of Vienna, Dept. of Functional and Evolutionary Ecology, Austria
[15]Oregon State University, Corvallis, Oregon, USA
[16]Ohio State University, Columbus, Ohio, USA
[17]Department of Biomolecular Medicine, Faculty of Medicine and Health Sciences, Ghent University, 9000 Ghent, Belgium
[18]VIB – UGent Center for Medical Biotechnology, VIB, 9000 Ghent, Belgium
[19]Department of Geophysical Sciences, University of Chicago, Chicago, Illinois, USA
[20]Rutgers University, Piscataway, New Jersey, USA
[+]Present address: University of Georgia, Department of Marine Sciences
* corresponding author, msaito@whoi.edu

*Submitted to Biogeosciences 12/27/2023*

*Revision for Biogeosciences 7/3/2024*

*Technical Correction for Biogeosciences 08/29/2024*

**Abstract**
Metaproteomics is an increasingly popular methodology that provides information regarding the
metabolic functions of specific microbial taxa and has potential for contributing to ocean ecology
and biogeochemical studies. A blinded multi-laboratory intercomparison was conducted to
assess comparability and reproducibility of taxonomic and functional results and their sensitivity
to methodological variables. Euphotic zone samples from the Bermuda Atlantic Time-Series
Study in the North Atlantic Ocean collected by *in situ* pumps and the AUV *Clio* were distributed
with a paired metagenome, and one-dimensional liquid chromatographic data dependent
acquisition mass spectrometry analyses was stipulated. Analysis of mass spectra from seven
laboratories through a common bioinformatic pipeline identified a shared set of 1056 proteins
from 1395 shared peptides constituents. Quantitative analyses showed good reproducibility:
pairwise regressions of spectral counts between laboratories yielded $R^2$ values averaged 0.62
+/- 0.11, and a Sørensen similarity analysis of the top 1,000 proteins revealed 70-80% similarity
between laboratory groups. Taxonomic and functional assignments showed good coherence
between technical replicates and different laboratories. A bioinformatic intercomparison study,
involving 10 laboratories using 8 software packages successfully identified thousands of
peptides within the complex metaproteomic datasets, demonstrating the utility of these software
tools for ocean metaproteomic research. Lessons learned and potential improvements in
methods were described. Future efforts could examine reproducibility in deeper metaproteomes,
examine accuracy in targeted absolute quantitation analyses, and develop standards for data
output formats to improve data interoperability. Together, these results demonstrate the
reproducibility of metaproteomic analyses and their suitability for microbial oceanography
research including integration into global scale ocean surveys and ocean biogeochemical
models.

## 1. Introduction

Microorganisms within the oceans are major contributors to global biogeochemical cycles, influencing the cycling of carbon, nitrogen, phosphorus, sulfur, iron, cobalt and other elements (Falkowski et al., 2008; Moran et al., 2022; Worden et al., 2015). 'Omic methodologies can provide an expansive window into these communities, with genomic approaches characterizing the diversity and potential metabolisms, and transcriptomic and proteomic methods providing insights into expression and function of that potential. Similar to other 'omics approaches, proteomics is increasingly being applied to natural ocean environments and the diverse microbial communities within them. When proteomics is applied to such mixed communities, it is generally referred to as metaproteomics (Wilmes and Bond, 2006). Metaproteomic samples contain an extraordinary level of complexity relative to single organism proteomes (at least 1-2 orders of magnitude) due to the simultaneous presence of many different organisms in widely varying abundances (McCain and Bertrand, 2019). In particular, ocean metaproteome samples are significantly more complex than the human proteome, the latter of which is itself considered to be a highly complex sample (Saito et al., 2019). Proteomics (including metaproteomics) provides a perspective distinct from other 'omics methods: as a direct measurement of cellular functions it can be used to examine the diversity of ecosystem biogeochemical capabilities, to determine the extent of specific nutrient stressors by measurement of transporters or regulatory systems, to determine cellular resource allocation strategies in-situ, estimate biomass contributions from specific microbial groups, and even to estimate potential enzyme activity (Bender et al., 2018; Bergauer et al., 2018; Cohen et al., 2021; Fuchsman et al., 2019; Georges et al., 2014; Hawley et al., 2014; Held et al., 2021; Leary et al., 2014; McCain et al., 2022; Mikan et al., 2020; Moore et al., 2012; Morris et al., 2010; Saito et al., 2020; Sowell et al., 2009; Williams et al., 2012). The functional perspective that metaproteomics allows is often complementary to metagenomic and metatranscriptomic analyses and can provide biological

insights that are distinct from organisms studied in the laboratory (Kleiner et al., 2019).
Moreover, the measurement of microbial proteins in environmental samples has improved
greatly in recent years, due to the advancements in nanospray-liquid chromatography and high-
resolution mass spectrometry approaches (Mueller and Pan, 2013; Ram et al., 2005; McIlvin
and Saito, 2021).

With increasing interest in the measurement of proteins and their biogeochemical

functions within the oceans, the metaproteomic data is beginning to establish itself as a valuable
research and monitoring tool. However, given rapid changes in technology and methods, as well
as the overall youth of the metaproteomic field, demonstrating the reproducibility and
robustness of metaproteomic measurements to microbial ecology and oceanographic
communities is an important goal. This is particularly true as applications for metaproteomics
expand in research and monitoring of the changing ocean environment, for example in global
scale efforts such as the developing BioGeoSCAPES program (www.biogeoscapes.org;
(Tagliabue, 2023)), which aims to characterize the ocean metabolism and nutrient cycles on a
changing planet. As a result, there is a pressing need to assess inter-laboratory consistency,
and to understand the impacts of sampling, extraction, mass spectrometry, and bioinformatic
analyses on the biological inferences that can be drawn from the data.

There have been efforts to conduct intercomparisons of metaproteomic analyses in both

biomedical and environmental sample types in recent years that provide precedent for this
study. A recent community best practice effort in ocean metaproteomics data-sharing also
identified major challenges in ocean metaproteomics research, including sampling, extraction,
sample analysis, bioinformatics pipelines, and data sharing, and conducted a quantitative
assessment of sample complexity in ocean metaproteome samples (Saito et al., 2019). A
previous benchmark study, driven by the Metaproteomics Initiative (Van Den Bossche et al.,
2021), was the "Critical Assessment of Metaproteome Investigation study" (CAMPI) that
employed a laboratory-assembled microbiome and human fecal microbiome sample to
successfully demonstrate reproducibility of results between laboratories. CAMPI found
robustness in results across datasets, while also observing variability in peptide identifications
largely attributed to sample preparation. This observation was consistent with prior findings on
single organism samples that determined >70% of the variability was due to sample processing,
rather than chromatography and mass spectrometry (Piehowski et al., 2013). Finally, the
Proteomics Informatics Group (iPRG) from the Association of Biomolecular Resources Facilities
(ABRF) conducted a study examining the influence of informatics pipelines on metaproteomics
analyses that found consistency among research groups in taxonomic attributions (Jagtap et al.,
2023), and previous research has demonstrated the impact of database choices on final
functional annotations and biological implications (Timmins-Schiffman et al., 2017).

Here we describe the results from the first ocean metaproteomic intercomparison. In this

study, environmental ocean samples were collected from the euphotic zone of the North Atlantic
Ocean and partitioned into subsamples and distributed to an international group of laboratories
(Fig. 1). The study was designed to examine inter-laboratory consistency rather than maximal
capabilities, stipulating one-dimensional chromatographic analyses from each laboratory (with
optional deeper analysis). Users were invited to use their preferred extraction, analytical, and
bioinformatic procedures. The effort focused on the data dependent analysis (DDA) methods,
also known as global proteomics where the targets are unknown and hence there is a discovery
element to the approach. DDA is currently common in ocean and other environmental and
biomedical metaproteomics, and its spectral abundance units of relative quantitation have been
shown to be reproducible in metaproteomics (Kleiner et al., 2017; Pietilä et al., 2022). Blinded
results were submitted, compared and discussed at a virtual community workshop in September
of 2021. An additional bioinformatic pipeline comparison study was also conducted where
participants were provided metaproteomic raw data and associated metagenomic sequence
database files and were encouraged to use the bioinformatic pipeline of their choice.
**2.  Methods**
*2.1 Sample Collection and Metadata*

Ocean metaproteome filter samples for the wet lab comparison (Figure 1) were collected

at the Bermuda Atlantic Time-series Study (31$^\circ$ 40'N 64$^\circ$ 10'W) on expedition BATS 348 on
June 16$^{th}$, 2018, between 01:00 and 05:00 am local time. *In situ* (underwater) large volume
filtration was conducted using submersible pumps to produce replicate biomass samples at a
single depth in the water column for intercomparisons. All filter subsamples are matched for
location, time, and depth. To collect the samples, two horizontal McLane pumps were clamped
together (Figure 1c) and attached at the same depth (80 m) with two filter heads (Mini-MULVS
design) on each pump and a flow meter downstream of each filter head. This depth was chosen
to correspond to a depth with abundant chlorophyll and photosynthetic organisms. Each filter
head contained a 142 mm diameter 0.2 µm pore-size Supor (Pall Inc.) filter with an upstream
142 mm diameter 3.0 µm pore-size Supor (Figure 1b, d). Only the 0.2 – 3.0 µm size fraction
was used in this study. The pumps were set to run for 240 min at 3 L per min. Volume filtered
was measured by three gauges on each pump, one downstream of each pump head, and one
on the total outflow (Table S2). Individual pump head gauges summed to the total gauge for
pump 1 (within 1 L; 447 L and 446.2 L), but deviated by 89 L on pump 2 (478 L and 388.9 L).
Given that the total gauge is further downstream, we report the pump head gauges as being
more accurate.

The pump heads were removed from the McLane pumps immediately upon retrieval,

decanted of excess seawater by vacuum, placed in coolers with ice packs, and brought into a
fabricated clean room environment aboard the ship. The 0.2 µm pore-size filters were cut in
eight equivalent pieces and frozen at -80°C in 2 mL cryovials, creating 16 samples per pump
that were co-collected temporally and in very close proximity (<1 m) to each other for a total of
32 samples used in this study (Figure 1d). The 3.0 µm pore-size filters are not included in this
study but are archived for future efforts. The sample naming scheme associated with the
different pumps and pump heads is described in Table S2. Note that pump 1A and 1B samples
accidentally had two 3.0 µm filters superimposed above the 0.2 µm filter, and 1B had a small
puncture in it, although neither of these seemed to affect the biomass collected, presumably the
puncture occurred after sampling was completed.

Samples for the bioinformatic component were collected by the autonomous underwater

vehicle *Clio*. The vehicle and its sampling characteristics were used as previously described
(Breier et al., 2020; Cohen et al., 2023). Specifically, samples  Ocean-8 and Ocean-11 were
also collected from the BATS station on R/V *Atlantic Explorer* expedition identifier AE1913 (also
described as BATS validation track BV55 32.75834º N 65.7374º W). The samples were
collected by autonomous underwater vehicle (AUV) *Clio* on June 19th 2019, dive Clio020, with
samples collected at 20 m (Ocean-11) and 120 m (Ocean-8) with 66.6 L and 92.6 L filtered,
respectively, used for this study. These depths were chosen to reflect the near surface (high-
light) and deep chlorophyll maximum (low-light) communities present in the stratified summer
conditions. These samples were analyzed by 1D DDA analysis using extraction and mass
spectrometry for laboratory 438 within their laboratory (Tables S5-S7). Sample metadata for
both arms of this intercomparison study and corresponding repository information is provided in
Table S3 and repository links are in the Data Availability Statement.
*2.2 Metagenomic Extraction, Sequencing, and Assembly*

A metagenomic (reference sequence) database was created for peptide to spectrum

matching  (PSMs) for the metaproteomic studies using a 1/8$^{th}$ sample split from the exact
sample used in the intercomparison as described above. Samples were shipped on dry ice to
the Naval Research Laboratory in Washington D.C. (USA), where DNA was extracted and
sequenced. Preserved filters were cut into smaller pieces using a sterile blade and placed into a
PowerBead tube with a mixture of zirconium beads and lysis buffer (CD1) from the Dneasy
PowerSoil Pro kit (Qiagen, Hilden Germany). The bead tube with filter sample was heated at
65°C for 10 min then placed on a vortex adapter and vortexed at maximum speed for 10 min.
After sample homogenization/lysis, the bead tube was centrifuged at 16 k $x$ g for 2 min. The
supernatant was transferred to a DNA LoBind tube and processed using the manufacturer's
recommendations. The purified DNA was further concentrated by adding 10 µL3 M NaCl and
100 µL cold 100% ethanol. The sample was incubated at -30°C for 1 hour, followed by
centrifugation at 16 k $x$ g for 10 min. The supernatant was removed and precipitated DNA was
air-dried and resuspended in 10 mM Tris. DNA concentration was quantified with the Qubit
dsDNA High Sensitivity assay (Thermo Fisher Scientific, Waltham, MA, USA) and DNA quality
was assessed using the NanoDrop (ThermoFisher) and gel electrophoresis. Processing controls
included reagent only and blank filter samples.
Sequencing libraries were created from purified sample DNA using the IonExpress Plus
gDNA Fragment Library Preparation kit (Thermo Fisher) for a 200 bp library insert size. No
amplification of the library was required as determined by qPCR using the Ion Library TaqMan
Quantitation Kit. A starting library concentration of 100 pM was used in template generation and
chip loading with the Ion 540 Kit on the Ion Chef instrument prior to single-end sequencing on
the S5 benchtop sequencer.
Sequencing used a mix of Ion Torrent and Oxford Nanopore sequencing and resulting
sequencing reads were assembled using SPAdes v. 3.13.1 with Python v. 3.6.8. Following
metagenome assembly, contigs smaller than 500 bases were discarded. Open reading frame
(ORF) calling was performed on contigs 500 bps or longer using Prodigal v. 2.6.3 (Hyatt et al.,
2010) run with metagenomic settings as well as MetaGeneMark by submitting to the
MetaGeneMark server (http://exon.gatech.edu/meta_gmhmmp.cgi) using GeneMark.hmm
prokaryotic program v. 3.25 on August 11, 2019. ORFs called from both programs were
combined and made non-redundant using in-house Python scripts that utilize BioPython v. 1.73.
Non-redundant ORFs were annotated using the sequence alignment program DIAMOND (v 0.9.29)
with the NCBI nr database (downloaded 12/17/2019). ORFs were also annotated with InterProScan
(v 5.29) and with GhostKOALA (Kanehisa et al., 2016) (submitted to server 1/2/2020). Taxonomy
lineages were generated by using the best DIAMOND (Buchfink et al., 2015) hit and pulling lineage
information from NCBI Taxonomy database using BioPython v. 1.73
*2.3 Proteomic methodologies: Extraction, instrumentation, and bioinformatics*

Some basic protocol stipulations were provided to study participants regarding analytical

conditions to set a uniformity of experimental design. While users were encouraged to use the
extraction method of their preference, constraints on chromatography and mass spectrometry
conditions were set, limiting the number of chromatographic dimensions to one (1D), the total
length of the chromatographic run, the amount of protein injected (as proteolytic digests), and a
single mass spectrometry injection rather than gas phase fraction approaches (Table S4). Each
laboratory group's specific approach is summarized in the supplemental methods, with
extraction in Table S5, and chromatography and mass spectrometry equipment and parameters
in Tables S6 and S7. While there are more sophisticated methods such as two-dimensional
(2D) chromatography and gas phase fractionations that have been demonstrated to provide
deeper metaproteomes (McIlvin and Saito, 2021), these often require specialized equipment
and/or additional instrument time. As a result, the study constraints were provided to ensure a
single simple method that all labs could utilize. Laboratories were invited to submit additional
data from more complex analytical setups if they first completed the 1D analyses.

*2.4 Compilation, analysis, and re-analysis of laboratory data submissions*

Results from individual laboratories' data submissions were analyzed in two ways as

shown in the flowchart of Figure 1a. First, submitted processed data reports (i.e. PSMs,
taxonomic, functional annotations) were compiled and interpreted. Second, raw data files (i.e.
spectra directly from instruments) from each group were put through a single bioinformatic
pipeline using SEQUEST HT/Percolator within Proteome Discoverer (Version 2.2.0.388,
Thermo Scientific) and Scaffold (Version 5.2.1, Proteome Software) to isolate variability
associated with bioinformatic processing. Note that Scaffold ignores the Percolator output from
Proteome Discoverer when re-running in Scaffold. This re-analysis (*single pipeline re-analysis*
hereon) allowed detailed cross-comparisons of laboratory practices to assess the influence of
the extraction and mass spectrometry components. Specific parameters of the latter included:
parent  of tolerances of 10ppm were used on all instruments (all Orbitraps) for fragments
tolerances of 0.02 Da or 0.6 Da were used for Orbitrap ms2 instruments and for ion trap ms2
instruments, respectively. Fixed and variable modifications of +57 on C (fixed), and +16 on M
and +42 on Peptide N-Terminal (variable) were used. Peptide and protein FDRs (false
discovery rates) were set to lower than 1.0% using a decoy database, with 1 minimum peptide
per protein, and the resulting peptide FDR was 0.1%. The database used for PSMs was
Intercal_ORFs_prodigal_metagenemark.fasta based on the metagenomic sequencing
described above with 197,824 protein entries. The re-analysis was conducted within Scaffold
using total spectral counts and allowing single peptides to be attributed to proteins. In addition to
the total number of protein identifications, the number of protein groups identified by Scaffold
was also provided. Each protein group represented proteins identified with identical peptides,
collapsed into a single protein entry with the highest probability and number of spectral counts.

*2.5 Data analysis methods*
Several analyses were conducted using data from the single pipeline re-analysis. First,
pairwise comparisons of protein identifications were conducted using spectral abundance
reports produced in Scaffold, and loaded, analyzed and visualized in MATLAB (MathWorks Inc).
Two-way (independent) linear regressions were conducted using the script linfit.m. $R^2$ on the
seven datasets were averaged and their standard deviation calculated for shared proteins in
each dataset. Second, a Sørensen similarity (Sørensen, 1948) was calculated where a matrix
was generated that consisted of the unique proteins or peptides identified across all technical
replicates from the various labs with the relative abundance per replicate (% contribution of
each protein/peptide per technical replicate total). The Bray-Curtis dissimilarity pairwise distance
was calculated on this matrix using Python and the SciPy library (v. 1.4.1, (Virtanen et al.,
2020)) and then 1 – Bray-Curtis dissimilarity was calculated across the matrix to generate the
Sørensen pairwise similarity across all replicates. The resulting similarities per replicate were
clustered and visualized using the clustermap function in the Seaborn library (v. 0.10.0,
(Waskom, 2021)). Third, shared peptides and proteins were visualized using Upset plots, using
the R package UpSetR (Conway et al., 2017) to determine the number of unique peptide
sequences and annotated proteins in intersecting sets between all labs, all permutations of lab
subsets, and all lab pairs.
*2.6. Bioinformatics Intercomparison Methods*
The methods used for the bioinformatics intercomparison study are described by each
laboratory using their unique three-digit identifier code. All laboratories used the metagenomic
database generated in the laboratory study (see Section 2.2). **Lab 109:** The raw files were
searched against the metagenomic database employing a 2 round search using PEAKS Studio
X. The initial database search was performed to focus the metagenomic database for protein
sequences with peptide sequence matches at 5% FDR. The focused database was further used
for a second round search, which allowed a parent mass error tolerance of 10.0 ppm and a
fragment mass error tolerance of 0.6 Da. The search considered up to 3 missed cleavages,
carbamidomethylation as fixed and methionine oxidation and N-terminal acetylation as variable
modifications. The cRAP protein sequences (http://ftp.thegpm.org/fasta/cRAP./) were included
as contaminant database. Finally, PSMs were filtered for 1% FDR and annotated with
taxonomic lineages (obtained from the metagenomic experiments). Non-unique peptide
matches were annotated with the LCA of the respective lineages.
**Lab 321:** SearchGUI (Galaxy Version 3.3.10.1) was used to search using multiple search
algorithms (X!Tandem, MS-GF+ and Comet). For each search algorithm, Precursor Tolerance
of 10.0 ppm, Fragment Ion Tolerance of 0.6 Da and trypsin was used as an enzyme for
proteolytic cleavage. Searches were performed allowing for two missed cleavages fixed
modification of Carbamidomethylation at cysteine and Variable Modifications of Acetylation of
protein N-term and Oxidation of Methionine. PeptideShaker (Version: 1.16.36) was used to filter
peptides with the length of 8-50 aas and a precursor m/z tolerance of 10.0 ppm. Detected
peptide-spectral matches, peptides and proteins were reported at 1% global FDR. All of the
analysis was performed within Galaxy platform.
**Lab 321:** MaxQuant (Galaxy version 1.6.17.0+galaxy3) was used to search the datasets. A
fixed modification of carbamidomethylation at cysteine and variable mmodifications of
acetylation of protein N-term and oxidation of methionine was applied along with allowing for
two missed cleavages. The detection peptides and proteins were reported at 1% FDR.
**Lab 362:** The raw files were converted using ThermoRawFileParserGUI (version 1.4.1) to peak
lists (.mgf files) using "native Thermo library peak picking" as the peak picking option and
"Ignore missing instrument properties" as the error option. The peak lists (.mgf files) obtained
from MS/MS spectra were identified using X! Tandem version X! Tandem (Vengeance version
2015.12.1) using SearchGUI version 4.1.0. Here, the parameters provided and suggested by
the study were used: tolerances of 10 ppm for MS1 and 0.6 Dalton for MS/MS; dynamic
modifications: oxidation of M, and acetyl on N-terminus; static modifications: carbamidomethyl
of C. Identification was conducted against a concatenated target/decoy database of the
provided database.
The X!Tandem files were used as input in MS²ReScore
(https://github.com/compomics/ms2rescore), a machine learning-based post-processing tool
that improves upon Percolator rescoring of peptide-to-spectrum matches (PSMs). Here, the
search engine-dependent features of Percolator were appended with MS2 peak intensity
features by comparing the PSM with the corresponding MS²PIP-predicted spectrum. All
reported MS²ReScore PSM identifications have a q-value < 0.01. No protein grouping algorithm
was applied, and all identified taxa and functions are extracted from the provided database.
**Lab 458:** The Proteome Discoverer 2.5 platform was used (SequestHT + Percolator (MPS)).
Fully tryptic peptides with a minimum length of 6 peptides and a maximum of 2 missed
cleavages were required.  Precursor Tolerance of 10.0 ppm, Fragment Ion Tolerance of 0.6 Da.
carbamidomethylation as fixed and methionine oxidation was set as a variable modification. Filtering
was performed at a 1% PSM- and peptide-level FDR.  The MaxQuant contaminant list was used as
a contaminant database.
**Lab 501:** We first appended the database with a set of common contaminants (Global
Proteome Machine Organization common Repository of Adventitious Proteins). Then, we used
MSGF+ (Kim and Pevzner, 2014) to match mass spectra with peptide sequences, with cysteine
carbamidomethylation as a fixed modification, and methionine oxidation, glutamine modified to
pyro-glutamic acid, deamidated asparagine, and deamidated glutamine, as variable
modifications. Peptides were searched for with a Target-Decoy approach, with a 1% false
discovery rate at the peptide spectrum match level. For spectral counts, we summed MS2
spectra that identified a peptide, and normalized all spectral counts to the total spectral counts
per sample. Proteins were quantified using the median spectral count for all proteotypic
peptides (those peptides which uniquely correspond to a protein), specifically using the
OpenMS tool ProteinQuantifier. This approach requires at least one proteotypic peptide, but if
more are identified, those peptides are also used for quantification.
**Lab 828:** The raw files were analyzed using Thermal proteome discover. MS/MS spectrums
were searched against provided database using SEQUEST-HT engine. MS/MS spectra
searches were performed as follows: precursor ion tolerance of 10.0 ppm; fragment ion
tolerance of 0.6 Da; carbamidomethyl cysteine was specified as fixed modification, whereas
oxidation (M), deamidation (N/Q), and N-terminal protein acetylation were set as variable
modifications. Trypsin was specified as the proteolytic enzyme, allowing for two missed
cleavages. Percolator-based scoring was chosen to improve the discrimination between correct
and incorrect spectrum identifications, learning from the results of a decoy and target database;
settings were as follows: maximum delta Cn, 0.05; strict false-discovery rate of 0.01 and
validation based on q values.
**Lab 902:** SEQUEST-HT was used within Proteome Discoverer 2.2 using the following settings:
maximum missed cleavage 2, minimum peptide length 6, maximum peptide length 122,
precursor mass tolerance 10ppm, fragment mass tolerance 0.6 Dalton; dynamic modifications:
M oxidation, acetyl on N-terminus; static modifications: C carbamidomethyl. Percolator PSM
validator (within Proteome Discoverer) with following settings: maximum Delta Cn 0.05, target
FDR strict 0.01, target FDR relaxed 0.05, validation based on PEP. Scaffold 5.0 used to analyze
Proteome Discoverer generated files with following settings: scoring system: prefiltered mode;
protein grouping: standard experiment wide protein grouping; protein threshold 1.0% FDR;
peptide threshold 0.1% FDR; minimum number of peptides 1.
**Lab 932:** Mass spectrometry data were transformed from Thermo RAW format (version 66) to
mzML and Mascot Generic (MGF) formats using ThermoRawFileParser (version 1.2.0,
Hulstaert et al., 2020). Experimental metadata were extracted from mass spectrometry data
using the MARMoSET program (Kiweler et al. 2019). Mascot Server (version 2.6.2, Matrix
Science, LTD) software performed peptide-spectrum matching between experimental data and
a reference sequence database. Reference sequences included a total of 197,824 predicted
protein-coding ORFs from a metagenome assembly. Peptides matching an in-house curated
inventory of contaminant protein sequences, mass standards, and proteolytic enzyme
sequences were removed from the results. Mascot search parameters included the following
settings: +10.0 ppm monoisotopic precursor mass tolerance; +0.6 Da monoisotopic fragment
ion tolerance; one fixed modification (+57 to C residues); two variable modifications (+16 to M
residues, +42 to peptide amino-termini); digestion enzyme trypsin; two missed cleavages;
peptide charges +2-+7; and instrument type: electrospray ionization coupled to fourier-transform
ion cyclotron resonance (ESI-FTICR). Mascot search results containing peptide-spectrum
matches (PSMs) were exported for downstream data analysis. Scaffold Q+S (version 4.8.9) was
used to validate MS/MS-based peptide- and protein-level peptide-spectrum matches (PSM) with
the Peptide Prophet algorithm. Mascot PSM data were imported into Scaffold Q+S with the
following settings specified: quantitative metric: spectrum counting; scoring system: use legacy
Peptide Prophet scoring (high mass accuracy); protein grouping: use standard experiment-wide
grouping; optional loading steps: pre-compute false discovery rate (FDR) thresholds; and, use
local gene ontology (GO) annotations (UniProt GO annotation data retrieved 25 JUN 2020).
Scaffold Q+S identification criteria were set at greater/equals >99.9% probability by the Peptide
Prophet algorithm (Keller et al. Anal. Chem. 2002.) and >99.9% probability by the Protein
Prophet algorithm (Nesvizhskii et al., Anal. Chem. 2003) with >2 peptides at the protein level.
**Lab 957:** MSFragger 3.3 searches were performed with FragPipe 16.0 and Philosopher 4.0.0. A
concatenated target/reverse database was searched with a 50 PPM precursor and 0.4 Da
fragment mass tolerance. Automatic mass calibration and parameter optimization was enabled
and precursor mass errors for up to +2 neutrons were considered. Peptide candidates were
generated from database protein sequences assuming tryptic digestion, allowing for up to one
missed cleavage. Peptides were required to have between 8-50 amino acids and range from
500 to 5000 m/z. Cysteines were assumed to be fully carbamidomethylated, and peptides were
searched considering variable n-terminal pyroglutamic acid formation and methionine oxidation.
PeptideProphet was used for FDR validation with the following default options: "--decoy probs",
"--ppm", "--accmass", "--nonparam", and "--expectscore", which allow for additional high-mass
accuracy analysis and non-parametric distribution fitting. ProteinProphet was used for protein-
level FDR validation with the following default option: "--maxppmdiff 2000000". Filtering was
performed using a 1% peptide-level and a 1% protein-level FDR threshold.
**3.  Results**
*3.1 Experimental Design*

This ocean metaproteomic intercomparison consisted of two major components: a

laboratory component, where independent labs processed identical ocean samples
simultaneously collected from the North Atlantic Ocean (Fig. 1a, see Section 2.1), and a
subsequent bioinformatic component. Participating institutions and persons at those institutions
are listed in Table S1, with all participants also listed as co-authors. Both arms of the study were
conducted under blinded conditions, where correspondence with participants was conducted by
an individual not involved in either study, and submitted results and data were anonymized prior
to sharing with the consortium. Within both arms of the study, participants were provided the
location of the study site and metadata about the sampling locations, time and depth at the
onset of the study. The laboratory study involved two biomass-laden filter slices collected from
the North Atlantic Ocean Bermuda Atlantic Time series Study site at 80m depth being sent to
each participating group for protein extraction, mass spectrometry, and bioinformatic analyses
(see Section 2.1). This depth was chosen to correspond to a depth with abundant chlorophyll
and associated photosynthetic organisms. The bioinformatic effort was independent of the
laboratory effort and involved the distribution and bioinformatic analysis of two metaproteomic
raw data files generated from samples also from the North Atlantic Ocean upper water column
BATS station (20m and 120m depths, see Section 2.1). These depth were chosen to reflect the
near surface (high-light) and deep chlorophyll maximum (low-light) communities present in the
stratified summer conditions.These files were distributed after labs had submitted their
laboratory extracted raw data files. The raw files from the bioinformatic study were distinct from
the samples used in the laboratory intercomparison study to avoid any biases from groups that
analyzed those samples previously. Submitted results from both components were anonymized
and assigned three-digit lab identifiers generated randomly with laboratory and bioinformatic
results from the same lab being assigned distinct identifiers.

We report results for two study components: Part 1 (Section 3.2) involves the data

generation intercomparison of distributed subsamples from the North Atlantic Ocean (Fig. 1;
Section 2.1). Part 2 (Section 3.3) was an bioinformatic intercomparison, where metaproteomic
raw files were shared with participants and processed results were submitted. Both components
were conducted as blinded studies, where each dataset was assigned a three digit randomly
generated identifier, with those identifiers used throughout the Results and Discussion.

*3.2 Mass Spectrometry Data Generation Intercomparison*

Nine laboratories submitted raw and processed datasets from the analysis of the

distributed Atlantic Ocean field samples (Table S1). The processed data submissions were
heterogeneous in output formats, statistical approaches, and parameter definitions. Because of
the challenges of comparing data derived from different types of statistical approaches used for
peptide and protein identification and inference, as well as the varying output formats from
various software packages, the user-generated data submissions were difficult to compile and
compare, resulting in variability in the number of identifications depending on the statistical
approaches and thresholds applied. These results are further discussed in the Supplemental
Section (Figure S1, Table S8). Despite these challenges, an average of 7142 +/- 2074 peptides
were identified across the pairwise comparisons (Figure S1c) representing 20% of the 35,715
total unique peptides detected across all labs. Together these findings implied a consistency of
peptide identifications across participants. The variability in proteome depth reflected the
combination of differing parameters employed by software and laboratory approaches.

To remove this variability associated with user-selected bioinformatic pipelines, a single

pipeline re-analysis of the submitted raw mass spectral data was conducted. Raw data files
were processed together within a single bioinformatic pipeline consisting of SEQUEST-HT,
Percolator, and Scaffold software and evaluated to a false discovery rate threshold of < 0.1% for
peptides and 1.0% for proteins  (see Section 2.4). Two datasets were found to have had issues
during extraction and analysis that affected the results in both processed and raw data (Labs
593 and 811; Table S8). Notably these two laboratories differed from the others in that they did
not use SDS as a protein solubilizing detergent (Table S5). This likely resulted in inefficient
extraction of the bacteria that dominated the sample biomass (e.g. picocyanobacteria and
*Pelagibacter*) embedded within the membrane filter slices. Further examination showed
polyethylene glycol contamination of one dataset (Lab 811) and low yield from sample
processing and extraction from the other (Lab 593). As a result, those datasets were not
included in the single pipeline re-analysis. The standardized pipeline included calculations of
shared peptides and proteins, quantitative comparisons, and consistency of taxonomic and
functional results.

The total number of peptide and protein identifications and PSMs in the single

bioinformatic pipeline analysis varied by laboratory (Table S9), with unique peptides ranging by
more than a factor of 3 from 3,354 to 16,500, and with 27,346 total unique peptides identified
across laboratories. This variability was likely due to different extraction, chromatographic, and
mass spectrometry hardware and parameters employed used by each laboratory, resulting in a
varying depth of metaproteomic results. Yet, as with the user-submitted results, there was
considerable overlap in identifications between all datasets. An intersection analysis found the
numerous shared peptides between all combinations of laboratories, with 1,395 peptides shared
between all seven laboratory datasets (Figure 2a). Laboratories with deeper proteomes shared
numerous peptides, for example the two laboratories with the most discovered unique peptides
shared ~3000 peptides between them, implying that shared peptides is a useful metric for
intercomparability. They also had the largest numbers of peptides that were not found by any
other labs (3617 and 2819, respectively). The fourth largest intersection size (1395) represented
the unique peptides discovered by all labs. Beyond that there were 12 different groupings of
peptides that were shared among at least four laboratories. Consistent with this, 3-way Venn
diagrams of labs 135, 209 and 438 had an intersection of 2398 peptides, labs 652, 729, and 774
shared 3016 peptides, and labs 127, 135, and 309 shared 2304 peptides (Figure 2d).

A similar analysis was conducted at the protein level, where the number of proteins

identified (see Section 2. Methods) identified 8,043 unique proteins in total across all
laboratories, with 1,056 proteins of those observed in all seven labs (see 7-way Venn diagram in
Figure 2c). Three-way Venn diagram comparisons among labs 135, 209 and 438 had an
intersection of 1,254 proteins, and labs 652, 729, and 774 shared 1,925 proteins (data not
shown).

Optional deeper metaproteome results were submitted by three laboratories using either

a long gradient of 12 hours or 2 dimensional chromatographic methods (Table S10). The
number of discovered peptide and protein identifications were higher in each case, with as
many as 18477 unique peptides and 7765 protein identifications from an online 2-dimensional
chromatographic analysis from a 5 $\mu$g single injection.

The mapping of identified peptides to protein sequences forms the basis for protein

identifications in the form of DDA bottom-up proteomics employed here. The relationship
between peptides and protein identification was explored in Figure 3 and found to be correlated
by two-way linear regression with $R^2$ values of 0.97 and 0.98 for total protein identifications and
protein groups, respectively. Together, the fact that there is a linear relationship between
peptides and proteins across all laboratories (including labs employing deeper methods) could
imply that the number of protein identifications has not begun to plateau and reached
'saturation', likely due to the immense biological diversity and abundance of lower abundance
peptides within these samples. This approach has some similarities to rarefaction curves used
in metagenomic sequencing to determine if the majority of species diversity has been sampled,
although in this case number of peptides used as a metric for sampling depth instead of
additional number of DNA sequencing samples typically used for rarefaction curves. This
indicated that with deeper depth of analysis by some laboratories, there was no fall off in the
increase in protein identifications that might be attributed to additional peptides mapping to
already discovered protein sequences. In addition, the 2D and long gradient additional analyses
conducted by several laboratories fell upon this line consistent with this "more peptides – more
proteins" observation, implying more room for improvements in depth of metaproteomic
analyses.

A quantitative analysis of spectral counts from the wet lab re-analysis showed broad

coherence among the seven laboratories. Pairwise comparisons of protein spectral counts were
conducted for each of the seven labs against the other six (visualized in a 7x7 matrix, with
duplicate comparisons removed (e.g., A vs B and B vs A)), where each data point reflects the
spectral counts for a protein shared between laboratories (Figure 4a). When a dataset was
compared with itself a unity line of datapoints was observed along the diagonal axis as
expected. Two-way linear regressions were conducted on each of these pairwise comparisons.
The slopes ranged from 0.33 to 5.5 (Figure S2), implying a varying dynamic range in spectral
counts across laboratories, likely due to variations in instrument parameterizations selected by
each laboratory, and consistent with the lack of normalization between laboratories. The
coefficient of determination $R^2$ values from 0.43 to 0.84 with an average of 0.63 +/- 0.11,
showing coherence among results for these large metaproteomic datasets (Figure 4b, Table
S12). To provide a sense of coherence of each laboratory to the others, the $R^2$ values of a lab
against the other six laboratories were averaged and the standard deviation calculated. All of
these average $R^2$ values were higher than 0.5, which showed overall quantitative consistency
despite the size and complexity of these datasets (Figure 4d).

A comparative taxonomic and functional analysis was also conducted using a single

bioinformatic pipeline (see metagenomic sequencing methods for annotation pipeline). Lowest
common ancestor (LCA) analysis of peptides identified from datasets from seven laboratories
showed consistent patterns of taxonomic distribution using the MetaTryp package (Figure 5a;
(Saunders et al., 2020). Cyanobacteria and alphaproteobacteria were the top two taxonomic
groups in all laboratory submissions, consistent with the abundant picocyanobacteria
*Prochlocococcus* and the heterotrophic bacterium *Pelagibacter ubique* known to be dominant
components of the Sargasso Sea ecosystem (Sowell et al., 2009; Malmstrom et al., 2010). For
example, *Prochlorococcus* is consistently present between $10^4$ and $10^5$ cells per milliliter in this
region and has been observed to contribute to carbon export from the euphotic zone (Casey et
al., 2007). *Pelagibacter* cells can also be in excess of $10^5$ cells per milliliter at the BATS North
Atlantic location (Carlson et al., 2009). These results are broadly similar to the representation of
phyla within the metagenome annotations, where Proteobacteria (including *Pelagibacter*) and
Cyanobacteria (including *Prochlorococcus* and *Synechococcus*) were major components,
although Bacteriodetes (including Flavobacteria) are more prevalent in the metagenome
annotations than in the metaproteome. Some differences may also be due to the incorporation
of protein abundances in Fig 5a, versus simple taxonomic attribution of non-redundant
assembled open reading frames in the metagenome analysis, as well as the use of multiple
sequencing platforms and gene calling algorithms (Section 2.2, Figure S4).

Similarly, KEGG Orthology group (KO) analysis of those datasets also showed highly

similar patterns of protein functional distributions across laboratories (Figure 5b). Notably the
PstS phosphate transporter protein from *Prochlorococcus* was the most abundant protein in all
datasets, consistent with observations of phosphorus stress in the North Atlantic oligotrophic
gyre and its biosynthesis in marine cyanobacteria (Scanlan et al., 1997; Coleman and Chisholm,
2010; Ustick et al., 2021). These findings demonstrate the reproducibility in the primary
functional and taxonomic conclusions from the metaproteome datasets. Finally, a Sørensen
similarity analysis of the 1,000 proteins with highest spectral counts revealed 70–80%
similarities between most laboratory groups in the data re-analysis (Figure 6). When conducted
on the full dataset with all peptides and proteins, the Sørensen similarity analyses showed
peptides had lower similarity than proteins, implying variability is ameliorated when aggregated
to the protein level (Figure S3).

*3.3. Bioinformatic Data Analysis Intercomparison*

Two metaproteomic raw files were provided to intercomparison participants and were

searched with each laboratory's preferred database searching bioinformatic pipeline. The
samples that generated the data for these files were collected by autonomous AUV *Clio* during
a single dive at the Bermuda Atlantic Time-series Study Station (Breier et al., 2020), and were
distinct from the samples associated with the laboratory intercomparison component. However,
they were also from the North Atlantic Ocean, allowing the same metagenomic database to be
used. This database was not collected simultaneously with the bioinformatics samples, so it was
not as representative as that used in the laboratory intercomparison. However, the BATS study
region is known to maintain similar major taxonomic composition throughout the year (e.g.,
*Prochlorococcus* and SAR11, see discussion in Section 3.2), hence enabling many protein
identifications. This bioinformatic study component was not launched until after the laboratory-
based intercomparison submission deadline to avoid influencing that part of the study by
sharing similar raw data. Samples were named Ocean 8 and Ocean 11 and were taken from
120 m and 20 m depths, respectively.

The bioinformatic intercomparison involved 10 laboratories utilizing 8 different software

pipelines including the PSM search engines: SEQUEST, X!Tandem, MaxQuant, MSGF+,
Mascot, MSFragger, and PEAKS (Table S11, see Methods Section 2.6). As with the user
supplied laboratory results, the results were challenging to compile due to different types of data
outputs, approaches used in protein inference, and statistical approaches applied within each
pipeline. Unique peptide discoveries served as a useful base unit of comparison that were less
subject to these comparison challenges. The number of peptides ranged from 1724 to 6369 in
Ocean 8 and 3019 to 8288 in Ocean 11 (Figure 7; Table S11). The differences in the number of
peptides was likely due to parameters used in software, for example, laboratory 932 had the
lowest number of peptides identified in both samples, but also used a highly stringent 99.9%
probability cutoff that likely influenced this result.

**4.  Discussion**
*4.1 Assessment of Ocean Metaproteomics Reproducibility*

Given the recent establishment of complex metaproteomic techniques, intercomparisons

are valuable in demonstrating their suitability for ocean ecological and biogeochemistry studies.
Synthesizing the results of the laboratory and mass spectrometry blinded intercomparison study
(Section 3.2) processed with a single bioinformatic pipeline (Section 2.4), we observed
consistent reproducibility with regards to three attributes of ocean metaproteomics analyses: 1)
the identity of discovered peptides and proteins (Fig. 2), 2) their relative quantitative
abundances (Figs. 4 and 6), and 3) the taxonomic and functional assignments within
intercompared samples (Fig 5). With over 1000 proteins identified across seven laboratories
and Sørensen similarity indexes typically higher than 70–80% (Fig. 6), the results demonstrate
consistent detection and quantitation of major proteins in the sample. These results provide
confidence that multiple laboratories can generate reproducible results describing the major
proteome composition of ocean microbiome samples to assess their functional and
biogeochemical activity .
While there is good agreement, this congregation of data allows further exploration of
the influence of methods on the results. In particular, as mentioned above the range of pairwise
comparisons had correlation coefficients ranging from 0.43 to 0.84, with most values falling
between 0.6 and 0.8 (Figure 4b and 4e; Table S12). This average of all correlation coefficients
described above (0.63 +/- 0.11) implied good reproducibility between laboratories in general.
We can explore what might have influenced the variability and lower range of coefficients. The
correlation coefficients of lab 209 had two of the three $R^2$ values below 0.499 in pairwise
comparisons (0.431 and 0.475), yet also had values that ranged from 0.61 to 0.70. Why would
this variability exist?  Lab 209 's methods differed from other labs in several ways: they used the
oldest and slowest instrument of the group (Thermo Orbitrap Elite), used CID instead of HCD for
fragmentation and rapid scan mode, and used an unusually long column of 200cm to
compensate for the older instrument (Table S6). As a result, lab 209 had the lowest number of
peptide (3354) and protein (1586) ID's of the seven labs (Table S9), which was several fold
lower than the lab with the highest number and reduced the number of shared peptides across
all laboratories. In pairwise comparisons, lab 209 had the lowest number of shared peptides at
an average of 1304. Interestingly however, lab 209 did not have the lowest number of total
spectral counts (63198), being close to the average (70843 +/- 27455), implying that more
abundant peptides were detected relative to rarer ones.
We initially suspected the lower $R^2$ values in pairwise comparisons with lab 209 may
have been related to comparisons to laboratories with similarly lesser peptide depth, but this
was not the case: the two lowest correlation coefficients for lab 209 were with laboratories 135
and 774 (the 0.431 and 0.475 values), the latter of which had the highest number of peptide
identifications. The answer for this difference in quantitative values maybe within the selection of
parameters used to sample peptide peaks: Both lab 135 and 774 used 60 second dynamic
exclusion, whereas the other 5 labs used dynamic exclusions between 10 and 30 seconds in
length (Table S7). This higher dynamic exclusion likely contributed to providing greater peptide
discovery depth, but at the cost of quantitative consistency with other laboratories, since this
parameter selects against repeat counting of abundant peaks and would reduce spectral counts
of the more abundant peptides that lab 209 was detecting. This result demonstrates the
influence of the mass spectrometer parameters in quantitative reproducibility when using global
proteomic DDA mode.
*4.2 Metrics in metaproteomics: Core versus rare "long tail" proteins*
While abundant proteins were consistently detected across seven laboratories'
submissions, there was substantial variability in the less abundant proteins (Fig. 2). This is
evident in Figure 8, where most of the 1063 proteins across seven laboratories in the re-
analysis were in the upper half of proteins when ranked by abundance. This simultaneous
consistency in abundant proteins and diversity in rare proteins (and their respective peptide
constituents) was likely a result of several factors. First, the intercomparison experimental
design stipulated 1D chromatography in order to provide straightforward comparisons that all
laboratories could accomplish. This contributed to study consistency, but also resulted in lesser
proteome depth compared to more elaborate methods such as 2D chromatography and gas
phase fractionation commonly in use. Second, the sample complexity of ocean metaproteomes
has been shown to be enormous, with a far greater number of low abundance peptides present
than HeLa human cell lines (Saito et al., 2019). The combined effect of these factors meant that,
while laboratories were able to detect abundant proteins consistently, there was considerable
stochasticity associated with the detection of less abundant peptides resulting in a long tail of
discovered lower abundance proteins.
Mass spectrometer settings such as dynamic exclusion, chromatography conditions, and
variation in sample preparation methods all likely contributed to this stochastic variability in rare
peptide detection among laboratories. Moreover, while all participating laboratories used
Thermo orbitrap mass spectrometers, there were seven variants of instrument model, including
some with Tribrid multiple detector capability (Table S6). While testing other mass spectrometry
platforms is of interest, this trend of community orbitrap usage in this study is consistent with the
broader proteomics community, where currently 9 of the top 10 instruments used in
ProteomeXchange consortium repository data submissions utilize orbitraps as of the manuscript
submission date (Deutsch et al., 2019). When conducting analysis of environmental samples,
choices can be made about instrument setup and parameters based on the scientific objectives,
for example if maximal proteome depth or robust quantitation while using a discovery approach
is desired. Future intercalibration efforts enlisting more sensitive metaproteomic methods such
as 2D-chromatography (McIlvin and Saito, 2021), more sensitive instruments (Stewart et al.,
2023), and other emerging methods can greatly improve detection and quantitation of rarer
proteins in metaproteomes, allowing exploration of the depths of state-of-the-art capabilities
rather than our present emphasis on interlaboratory consistency. Moreover, the development
and adoption of best practices in sample collection, extraction, chromatographic separation,
mass spectrometry analyses, and bioinformatic approaches will contribute to interlaboratory
consistency.
*4.3* Despite the inter-laboratory variability in the detected sets of rarer peptides and proteins, we

interpret these to be largely robust identifications. The stringent 0.1% peptide-level FDR

threshold we use here is determined by scoring decoys: reverse sequenced peptides that

are not in our samples. Peptide assignments to these decoys model the score distribution of

all incorrect peptide-spectrum matches (PSMs) in our study such that FDRs can be

estimated in an unbiased way for each laboratory. However, these estimates are

complicated by subtle sequence diversity within a population's proteome, which is typically

not considered by proteomics software designed to analyze single species (Schiebenhoefer

et al., 2019). This diversity within metaproteomic samples results in the presence of highly
similar peptides with nearly identical precursor masses that produce many of the same b-
and y-ions, and this similarity is not well modeled by decoy peptides. The influence of
microdiversity on metaproteomics FDR estimation using strain-specific proteogenomic
databases is an important area of future exploration (Wilmes et al., 2008).*Bioinformatics*
*Intercomparison Assessment*
The discovery of peptide constituents of proteins within a complex ocean metaproteomic
matrix was successful across all software packages tested (Figure 7), where the metric for
success is a comparable number of peptide identifications. This is a notable finding due to the
highly complex mass spectra, large number of chimeric peaks present (Saito et al., 2019), and
large database sizes involved in ocean metaproteomes. To our knowledge, some of these
software packages had not yet been applied to ocean metaproteomes. There was also
variability associated with the stringency of statistical parameters employed, which points to the
challenges in assembling datasets from multiple laboratories with different depth of proteome
identifications.
Despite the success of this intercomparison component across software packages, there
is likely considerable room for improvement in the future. As mentioned previously, ocean
samples are highly complex and there are likely additional peptides that remain unidentified
using current technology, due to low intensity peaks and co-elution with other peptides resulting
in the chimeric spectra. Significant improvements in depth of analysis can be achieved through
increased chromatographic sample separation and optimized (or alternative) mass spectrometry
data acquisition strategies. Yet there is room for bioinformatic improvements as well: most DDA
database searching algorithms are unable to identify multiple peptides within a single
fragmentation spectrum. Moreover, when in DDA collection mode mass spectrometry software
typically does not isolate and fragment peptides that cannot be assigned a charge state, which
is a common occurrence for the low abundance peaks within ocean samples. As a result, there
is considerable room for improvements in bioinformatic pipelines to discover additional peptides.
Although the application of data independent approaches (DIA) to oceanographic
metaproteomics analysis has been limited (e.g. Morris et al., 2010), the systematic nature of ion
selection and fragmentation allows for a greater number of low abundant peptides to be
quantified when enough ions can be isolated to produce robust MS2 spectra.,.
*4.4 Lessons Learned and Future Efforts in Ocean Metaproteomic Intercomparisons and*

*Intercalibrations*

As the first interlaboratory ocean metaproteomics study, we chose to describe this study

as an intercomparison rather than an intercalibration and it served as a vehicle with which to
assess the extent of reproducibility. There were several lessons learned that can be
summarized here. These include the efficacy of a SDS detergent and heat treatment in lysing
and solubilizing marine microbial cells embedded on membrane filters, the significant problem
of data intercomparability between PSM software outputs and need for data output
standardization, and the influence of different hardware capabilities (Orbitrap generation) and
their parameter settings such as dynamic exclusion on proteome depth and quantitative
comparisons of spectral counts. The development of best practices associated with sample
collection, extraction, and analysis would be valuable, while also encouraging methodological
improvements and backward compatibility through the use of reference samples.

Future intercalibration efforts could aim to further assess and improve upon the level of

accuracy, reproducibility, and standardization of ocean metaproteome measurements.  In
particular, alternative modes of data collection and quantitation could also be tested in future
interlaboratory comparisons, including parallel reaction monitoring mode (PRM), multiple
reaction monitoring mode (MRM), quantification using isotopic labeling or tagging, and DIA
methods. PRM and MRM methods allow sensitive targeted measurements of absolute
quantities of peptides (e.g. copies per liter of seawater in the ocean context). As many 'omics
methodologies applied in environmental settings operate in relative abundance modes, adding
the ability to measure absolute quantities would be particularly valuable for comparisons of
environments across space and time. Targeted metaproteomic methods have been deployed in
marine studies using stable isotope labeled peptides for calibration, achieving femtomoles per
liter of seawater estimates of transporters, regulatory proteins, and enzymes (Saito et al., 2020;
Bertrand et al., 2013; Saito et al., 2014, 2015; Joy-Warren et al., 2022; Wu et al., 2019). These
methods are not yet widely adopted, but with growing interest could be deployed to other
laboratories and incorporated into future iterations of intercomparison and intercalibration
studies. DIA also has great potential in ocean metaproteome studies and is increasingly being
deployed in laboratory and field studies of marine systems. Similar to this DDA intercomparison,
the methodological and bioinformatic challenges of DIA could be explored during
intercomparisons of analyses of ocean samples. Finally, as mentioned above, all participants of
this study used orbitrap mass spectrometers for DDA submissions, but new instrumentation
such as trapped ion mobility spectrometry time of flight mass spectrometers (timsTOF) may be
applied to ocean metaproteome analyses and would be important to intercompare with orbitrap
platforms.

As noted above, there were also challenges in collating and comparing data outputs

from various software, as well as variation in how those programs conducted protein inference.
For example, peptide-level data from different research groups were reported as either
unmodified peptide sequences or as various peptide analytes (where modifications and charges
states were included with the peptide sequence), making compilation of peptide reports difficult.
Similarly, at the protein level reported proteins could be counted either before or after protein
grouping, e.g. applying Occam's-razor logic to peptide groupings into proteins – the former
reflecting the set of all proteins in the database that could be in the sample, the latter the
minimum set required to explain the peptide data. Such issues will also contribute to challenges
in integration and assembly of data from different laboratories for large ocean datasets. While
best practices for metadata and data types have been described by the community that include
specific attributes important for environmental and ocean samples such as geospatial location
and sample collection information (Saito et al., 2019) similar to the metadata standard recently
put forward in the human proteome field (Dai et al., 2021), this study also demonstrated that
there is  a need for standardization of data output formats for metaproteomic results.,.
*4.5 Metaproteomics in Global Ocean Surveys*

Understanding how the oceans are responding to the rapid changes driven by human

alteration of ecosystems is a high priority. Ocean and environmental sciences have a long
history of chemical measurements that are critical to assessing ecosystems and climatic
change. Such measurements have been straightforward for discrete measurements, such as
temperature, pH, chlorophyll, phosphate, dissolved iron and numerous other variables. When
collected over large spatial (ocean basin) or temporal (seasonal or decadal spans) scales, these
datasets have been powerful in identifying major (both cyclical and secular) changes. 'Omics'
measurements represent a more complex data type where each discrete sample can generate
thousands (if not more) of units of information. This study demonstrates the power and potential
for collaborative metaproteomics studies to identify key functional molecules and relate them to
their taxonomic microbial sources within the microbiome from multiple lab groups. Moreover,
multi-lab metaproteomics results in vastly enhanced identification of low abundance proteins
that are not identified by all research groups. Such low abundance proteins can be more likely
to change in abundance with changing environmental conditions and nutrient limitations,
resulting in a more nuanced and richer investigation of marine microbial ecology and
biogeochemistry with collaborative metaproteomics research. The implementation of such
voluminous data is beginning to be applied on larger scales and holds great promise in
improving not only our understanding of the functioning of the current system, but also the way
we assess how environments are changing with continued human perturbations.

Intercomparison and intercalibration are critical activities to undertake in order to allow

comparison of 'omics results across time and space dimensions. With major programs
underway and being envisioned such as the BioGEOTRACES, AtlantECO, Bio-GO-SHIP, and
BioGeoSCAPES efforts, the imperative for such intercalibration has grown and the need for best
practices is urgent. This Ocean Metaproteomic Intercomparison study is a valuable step in
assessing metaproteomic capabilities across a number of international laboratories,
demonstrating a clear consistency in measurement capability, while also pointing to the
potential for continued community development of metaproteomic capacity and technology.

*Author Contributions*: MAS and MRM obtained OCB workshop support and drafted the
experimental design with feedback from BN, MJ, and DL acting as the Advisory Committee. SC,
JH, DL, GJV, and JKS conducted the metagenomic analyses and assembly. JKS, MAS, MMB,
MRM, and RM conducted data analysis and visualization. MRM, MAS, JAB, MVJ, and RJ
conducted sample collection and/or AUV Clio operations. MAS, JKS, MRM, EMB, SC, JRC, TG,
JH, RLH, PJ, MJ, RK, HK, DL, JSPM, EM, SM, DMM, JN, BN, JJ, MD, GJH, RG, RM, BLN, MP,
SP, AR, ER, BS, TVDB, JRW, HZ, and ZZ contributed mass spectrometry data and/or
bioinformatics data for the intercomparison. JKS anonymized data submissions and conducted
follow-up correspondence about methods. The manuscript was drafted by MAS and all authors
contributed to the writing and editing.

*Data and Code Availability:* The raw files, metagenome database
(Intercal_ORFs_prodigal_metagenemark.fasta), and associated annotations
(Intercal_assembly_annotations.csv) for this project summarized in Table S3 are available at
ProteomeXchange and PRIDE repository with the dataset identifier PXD043218
(https://www.ebi.ac.uk/pride/archive/projects/PXD043218) and PXD044234
(https://www.ebi.ac.uk/pride/archive/projects/PXD044234). Co-located information about these
datasets are available at the Biological and Chemical Data Management Office under project
765945 (https://www.bco-dmo.org/project/765945) and at the BATS page (https://www.bco-
dmo.org/project/2124). The metagenomic reads are listed under Bioproject Accession:
PRJNA932835; SRA submission: SUB12819843, available at link:
https://www.ncbi.nlm.nih.gov/bioproject/PRJNA932835. The code for upset visualization is
available at: https://maggimars.github.io/protein/PeptideUpSetR.html.

*Competing Interests* - The authors declare no competing financial interests.
*Supplemental Materials* - Methods for the bioinformatic intercomparison study are available in
the Supplemental Methods. Supplemental Informational is available as Tables S1-S11, and
Figures S1-S3.
*Acknowledgements* - This manuscript is a product of the sustained efforts of a small group
activity supported by the Ocean Carbon & Biogeochemistry (OCB) Project Office (NSF OCE-
1850983 and NASA NNX17AB17G), based on a proposal written by M.A.S. and M.R.M. The
research expedition where samples were collected was supported by the NSF Biological
Oceanography and Chemical Oceanography. We also thank the R/V *Atlantic Explorer* and the
Bermuda Atlantic Time-series Study team for assistance at sea. AUV Clio sample collection was
supported by NSF OCE 1658030 and 1924554. Analyses by participating laboratories
acknowledge support from: NSERC Discovery Grant RGPIN-2015-05009 and Simons
Foundation Grant 504183 to E.M.B, the Austrian Science Fund (FWF) DEPOCA (project
number AP3558721) to G.J.H., Simons Foundation grant 402971 to J.R.W., National Institute of
Health 1R21ES034337-01 to B.L.N.,  the Norwegian Centennial Chair Program at the University
of Minnesota for funding to PDJ, SM, and TJG, NIH R01 GM135709, NSF OCE-1924554, OCE-
2019589 and Simons Foundation Grant 1038971 to M.A.S. Identification of certain commercial
equipment, instruments, software, or materials does not imply recommendation or endorsement
by the National Institute of Standards and Technology, nor does it imply that the products
identified are necessarily the best available for the purpose. We thank Magnus Palmblad, John
Kucklick, and an anonymous reviewer for comments on the pre-submission version of the
manuscript. We also thank two anonymous reviewers for their constructive comments during
manuscript review.

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

**Figure Captions**
**Figure 1.** Ocean metaproteomics intercomparison experimental design and sample collection.
a) The laboratory component (left) consisted of collection of field samples, 1-dimensional (1D)
chromatographic separation followed by data dependent analysis (DDA) uniformly employing
orbitrap mass spectrometers analyses by participating laboratories and submission of raw and
processed data. The bioinformatic (right) component consisted of distribution of two 1D-DDA
files, peptide-to-spectrum matching (PSMs), and submission and compilation of results. b) Size-
fractionated sample collection on 3.0 $\mu$m pore-size filter followed by a 0.2 $\mu$m pore-size Supor
filter, and the 0.2–3.0 $\mu$m size fraction was used for the intercomparison study. c) Two horizontal
*in-situ* McLane pumps were bracketed together with two Mini-MULVS filter head units each and
deployment on synthetic line. d) The four 142 mm filters were sliced into eighths (inset) and two
slices were distributed to each participating laboratory.

**Figure 2.** Shared peptides and proteins between laboratory groups using laboratory
submissions processed through a single bioinformatics re-analysis pipeline. a) Total number of
discovered unique peptides varied by more than three-fold among seven laboratory groups
(horizontal bars) due to varying extraction and analytical schemes (FDR 0.1%). The number of
intersections between datasets across all seven datasets was 1395 (fourth blue bar from left),
and various sets of intersections of peptides were observed amongst the data. b) Total number
of discovered proteins (FDR < 1%) varied more than four-fold from 1586 to 6221 among labs
(horizontal bars). Intersections between datasets across all seven laboratories was 1056, with
various sets of intersections of proteins observed, similar to the peptides. c) 7-way Venn
diagrams of shared unique peptides between laboratories showed 1056 shared peptides
between the 7 laboratories. d) 3-way Venn diagrams showed 2398, 2304, and 3016 shared
unique peptides between laboratories.

**Figure 3.** Comparison of unique peptides and discovered proteins. Comparison as total protein
identifications and protein groups from the single pipeline re-analysis based on submissions
from 9 laboratories. Increasing sample depth is linear with mapping to proteins, ($R^2$ of 0.97 and
0.98 for total protein IDs and protein groups, respectively, with slopes of 0.37 and 33) implying
that additional peptide discovery leads to proportionally more protein discovery, and that protein
discovery has not yet begun to saturate with more peptides mapping to each protein. Because
simple 1D analyses were stipulated in the intercomparison experimental design, peptide and
protein discovery was correspondingly limited in depth.

**Figure 4.** Quantitative comparison of intercomparison results. a) Pairwise comparisons of
quantitative abundance across six laboratories in units of spectral counts (comparisons with
itself show unison diagonals). b) $R^2$ values from pairwise linear regressions. d) Total proteins
identified in each laboratory. d) Average of each laboratory's $R^2$ values from pairwise regression
with the other six laboratories (error bars are standard deviation). In all cases average $R^2$ values
are higher than 0.5. e) Occurrences of $R^2$ values in pairwise comparisons spanning 0.4 to 0.9.
Potential causes of this range are outlined in the Discussion section.

**Figure 5.** Taxonomic and functional analysis of metaproteomic intercomparison. a) Percent
spectral counts by taxonomy was similar across laboratories and technical replicates within
laboratories. The sample was dominated by cyanobacteria and alphaproteobacteria,
corresponding primarily to *Prochlorococcus* and *Pelagibacter*, respectively. b) Percent spectral
counts per Kegg Ontology group showed the functional diversity of the sample.

**Figure 6.** Quantitative Sørensen similarity analysis. Analysis of top 1000 proteins (~75% of all
proteins) showed 70–80% similarity between most laboratory groups. Technical triplicates for
each laboratory group are shown.

**Figure 7.** Intercomparison of bioinformatic pipelines among laboratories. Unique peptide
identifications for sample Ocean 8 from 120m depth (a) and Ocean 11 from 20m depth (b), both
from the North Atlantic Ocean (Table S3), using a variety of pipelines and PSM algorithms.

**Figure 8.** Variability in discovered proteins between laboratories occurs in lower abundance
proteins. Top 7 panels: Abundance of proteins as percentage of total protein spectral counts
within each laboratory (y-axis is percentage), with proteins on the x-axis shown by ranked
abundance as the sum of spectral counts across all laboratories. Almost all proteins fall below
1% of spectral counts within the sample, and deeper proteomes have lower percentages due to
sharing of percent spectral counts across more discovered proteins. Bottom panel: Shared
proteins were found early within the long-tail of discovered proteins: the 1056 proteins shared
between all laboratory groups are almost all found to the left side indicating their higher
abundance in all seven datasets. Scale is binary in the seventh panel indicating presence in 7
labs or not.


Figure 1.



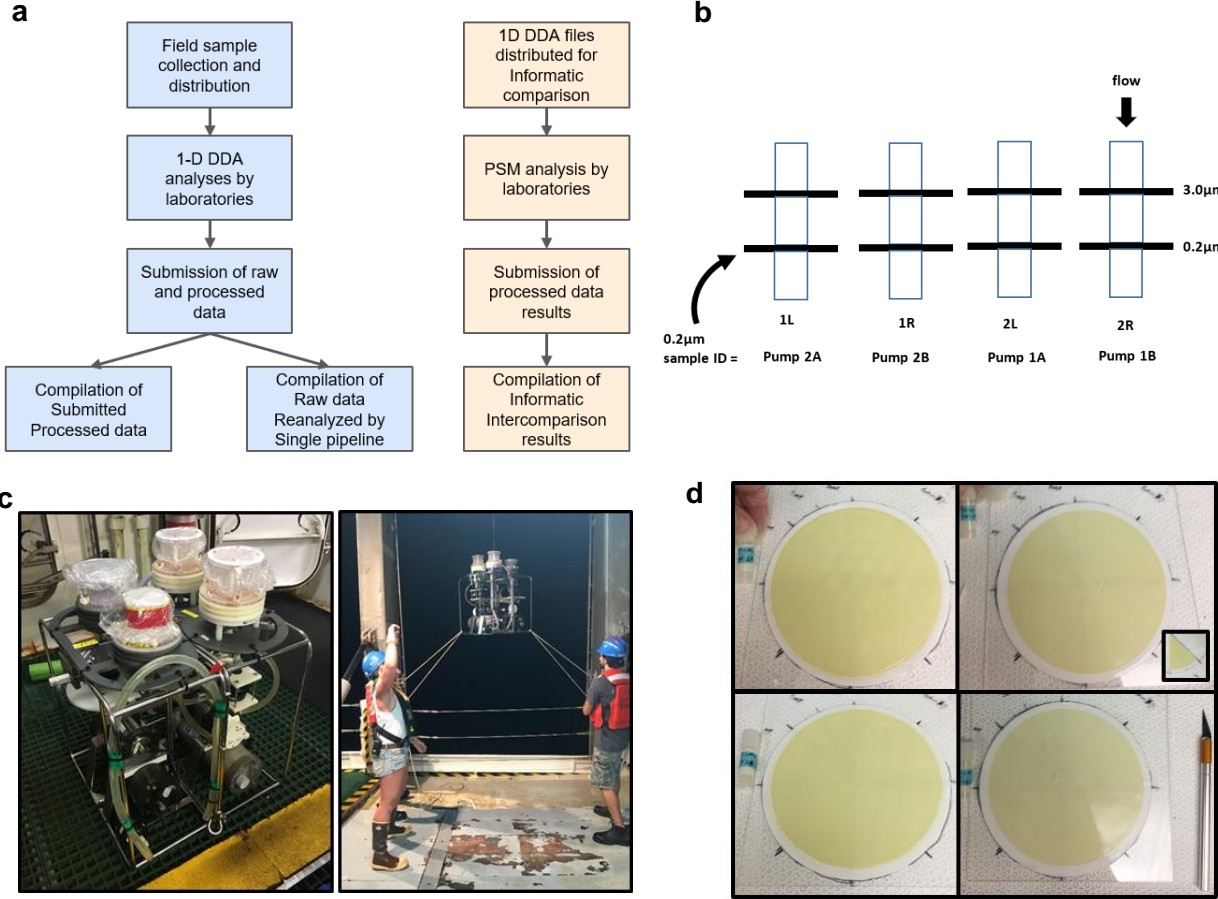

Figure 2.
**a**


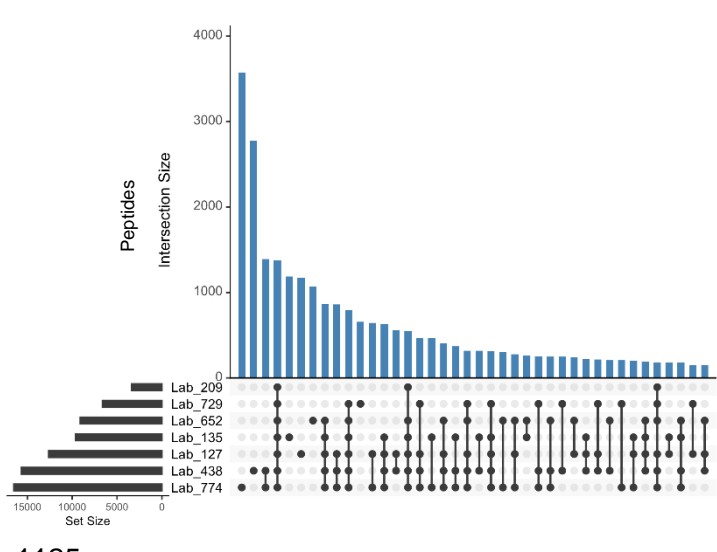

**c**

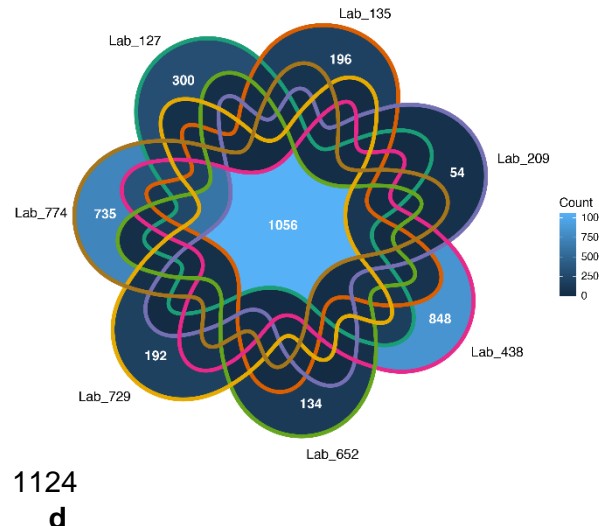




**d**

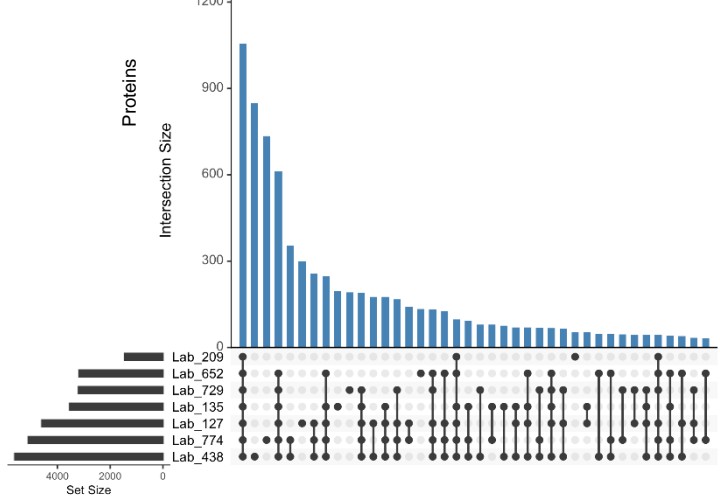

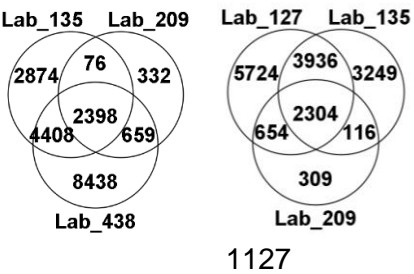


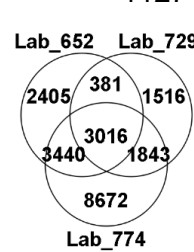




Figure 3








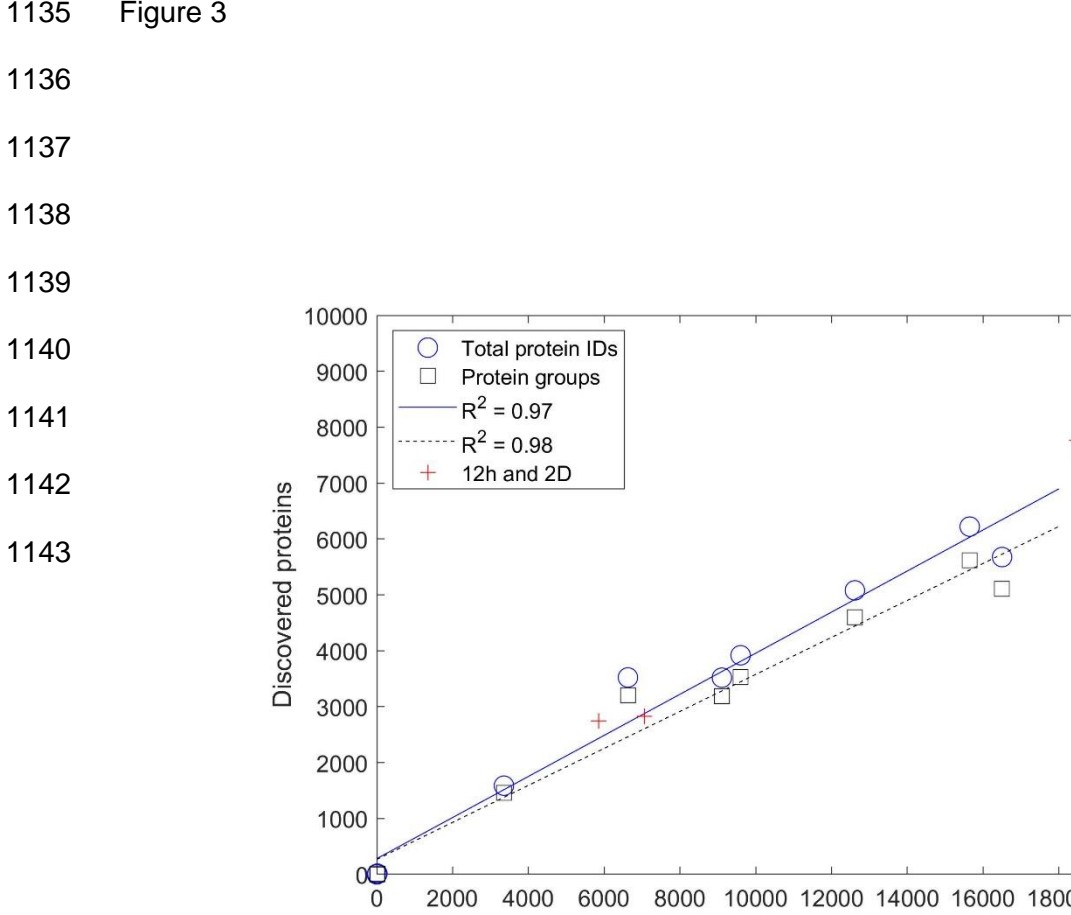

Figure 4.

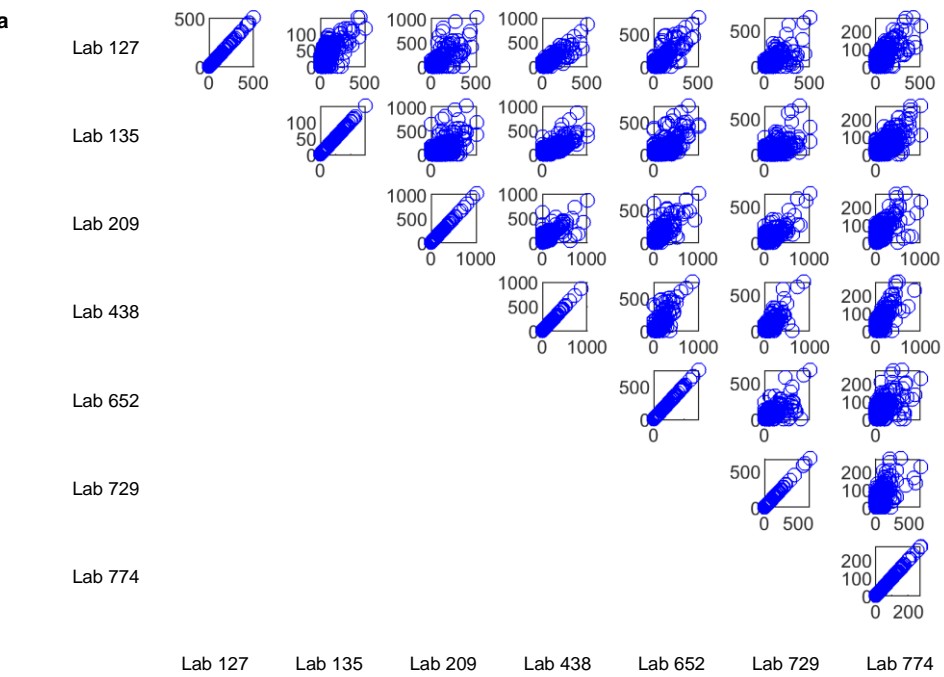

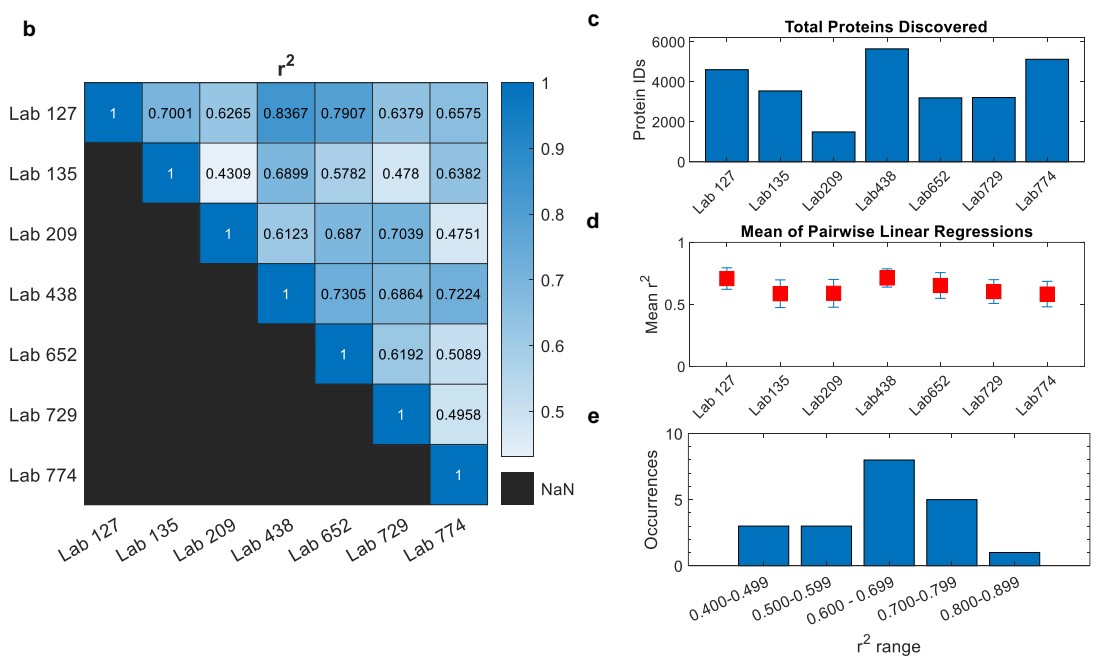

Figure 5.












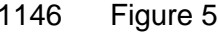

Figure 6.

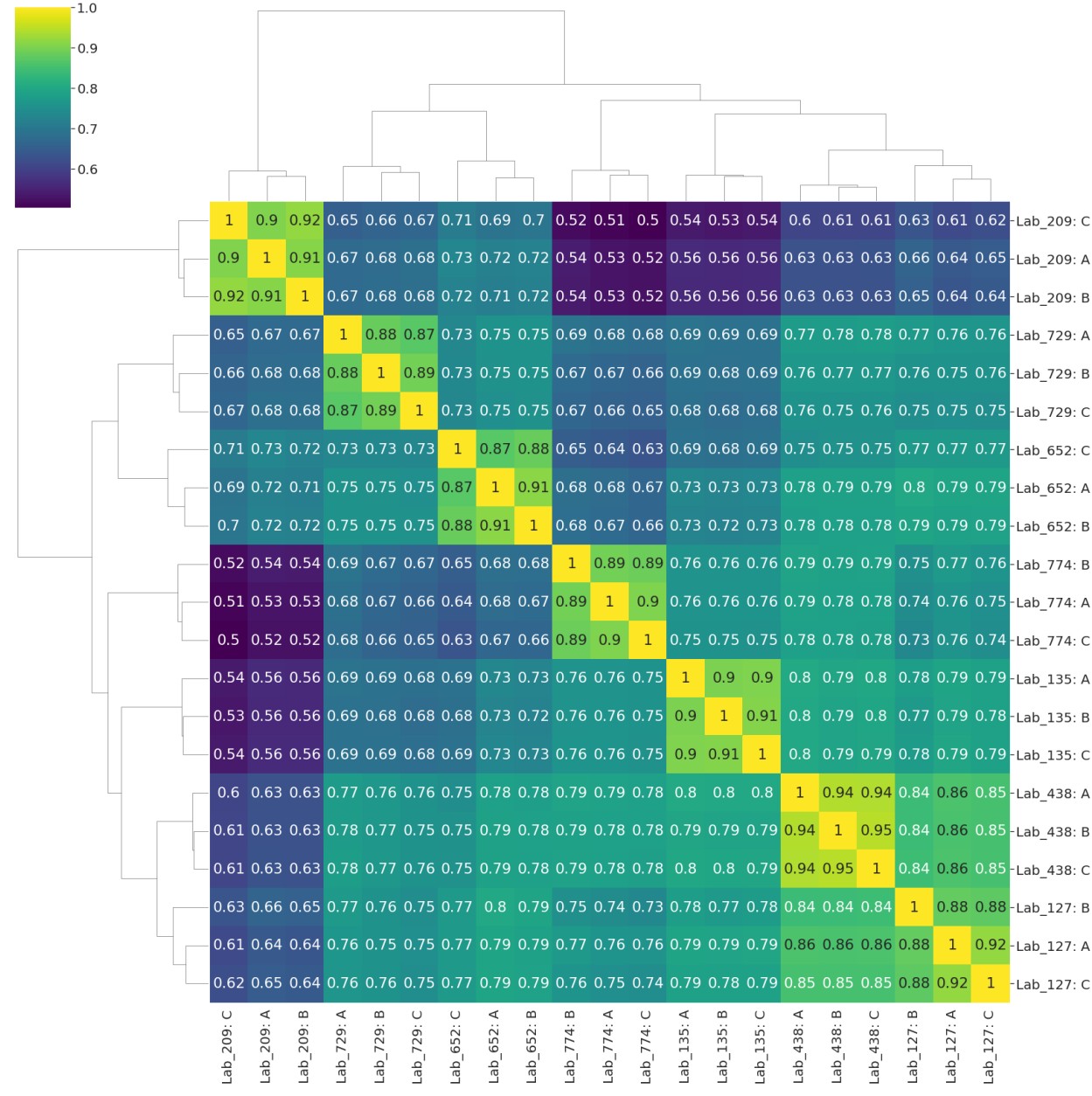







Figure 7.



a                              b

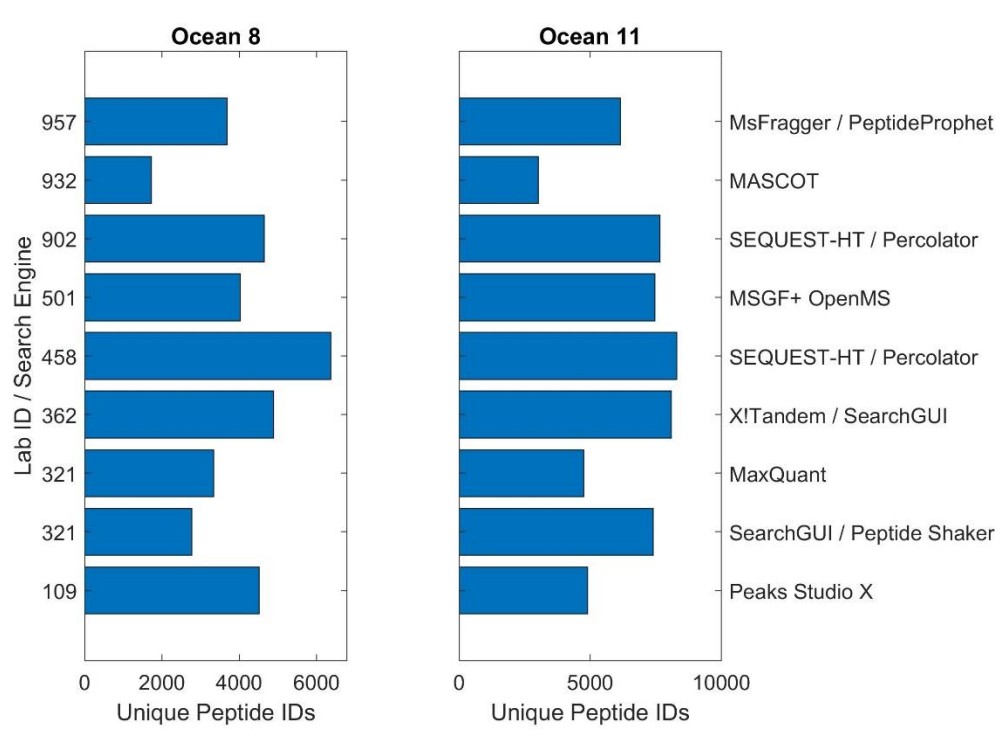



Figure 8.

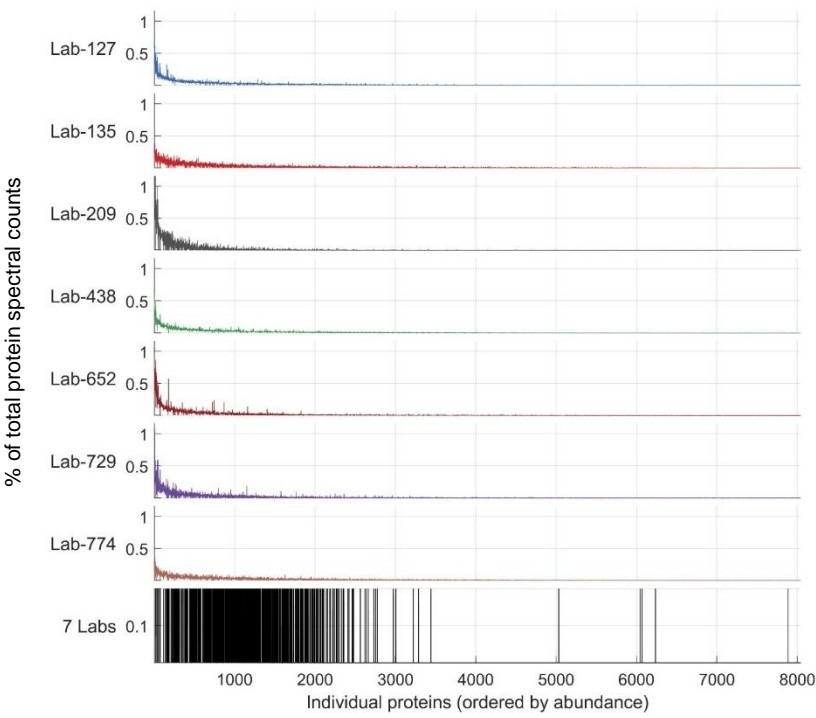