# Peer review of "Results from a Multi-Laboratory Ocean Metaproteomic Intercomparison"

_EGUsphere, 2023_

## Author Response (AR1)

**Reviewer Comments 1**

RC1: 'Comment on egusphere-2023-3148', Anonymous Referee #1, 26 Feb 2024

The authors describe the results of an intercomparison study of ocean discovery metaproteomics data, in which 9 labs participated (representing a very large fraction of the current ocean proteomics scientific community). This is important and timely work for the ocean omics community, which is embarking on multiple, large, coordinated efforts to collect data about ocean metabolism. The authors detail that, despite challenges in comparing data outputs from multiple labs and despite labs differing in almost all steps of proteomic analysis, the big picture microbial community composition and functions are largely consistent across lab groups. There is also evidence that a collaborative metaproteomics approach (i.e. data collected from multiple labs) can assist in the discovery of rarer proteins. This is encouraging news.

The intercomparison is focused on 1D DDA, discovery metaproteomics methods implemented on Orbitrap instruments. This is the most common and accessible proteomics approach available to ocean researchers (in part due to the utility of Orbitrap instruments for other ocean disciplines such as geochemistry, meaning they tend to be accessible in ocean science labs and departments). While developments in 2D fractionation, DIA methods, targeted methods, and non-Orbitrap mass analyzers are exciting and will no-doubt influence the future development of ocean metaproteomics, the authors approach of focusing with current popular methods makes sense at this time.

I understand that this kind of study is challenging, the field is young, and that the authors do not wish to be prescriptive. However, one clear message was the difficulty of comparing metaproteomics data across labs, and this is striking even though the labs were provided with the same samples and sequence database. This necessitated a re-analysis by the main arbiters of the intercomparison. I wonder if the authors could include some more details about what *could* have helped them in that comparison. For instance, would it have been beneficial to have multiple results from different types of searches (e.g. with and without protein groupings, with and without razor peptides?) If so, this would mean that with relatively little effort in re-analyzing their own data multiple ways, metaproteomics researchers could make their data more useful and re-usable.

*We appreciate these constructive comments. We have modified the last section of the manuscript to include a Lessons Learned component with the following new text: "There were several lessons learned that can be summarized here. These include the efficacy of a SDS detergent and heat treatment in lysing and solubilizing marine microbial cells embedded on membrane filters, the significant problem of data intercomparability between PSM software outputs and need for data output standardization, and the influence of different hardware capabilities (Orbitrap generation) and their parameter settings such as dynamic exclusion on proteome depth and quantitative comparisons of spectral counts. The development of best practices associated with sample collection, extraction, and analysis would be valuable, while also encouraging methodological improvements and backward compatibility through the use of reference samples."*

The rest of my comments focus on methodological details and opportunities for further discussion.

**General comments**

I'm finding the some figures to be fuzzy and in most cases the figure fonts to be quite small. This could be because of formatting in the preprint but I suggest to check this in future versions.

*We have checked the figures, replacing some, and will work with the published during proofs to ensure high resolution in the final versions.*

I'm curious to know how similar the relative abundances of the major organisms (cyanobacteria, alphaproteobacterial, and gammaproteobacterial) are in the metaproteomic data versus the matched metagenomes. Metagenome read recruiting is commonly used to assess microbial community composition, but there is evidence that metaproteomics can also provide good or better information.

*We have added an analysis to the supplemental information showing the fraction of genes annotated to each taxonomic group. The following text was added to the manuscript: "These results are broadly similar to the representation of phyla within the metagenome annotations, where Proteobacteria (including Pelagibacter) and Cyanobacteria (including Prochlorococcus and Synechococcus) were major components, although Bacteriodetes (including Flavobacteria) are more prevalent in the metagenome annotations than in the metaproteome. Some differences may also be due to the incorporation of protein abundances in Fig 5a, versus simple taxonomic attribution of non-redundant assembled open reading frames in the metagenome analysis, as well as the use of multiple sequencing platforms and gene calling algorithms (Section 2.2, Figure S4)."*

**Figure S4.** *Phylum distribution within metagenomic annotations with sum of each taxa as a fraction of all annotated genes.*

[Figure]

It would be useful to point out the special significance of the depths that were sampled and whether there were expected differences in microbial community composition among them (i.e. were the 80 and 120m samples the DCM?)

*The following text has been added to the wet-lab methods: "This depth was chosen to correspond to a depth with abundant chlorophyll and associated photosynthetic organisms."*

*And for the informatics samples: "These depth were chosen to reflect the near surface (high-light) and deep chlorophyll maximum (low-light) communities present in the stratified summer conditions."*

The informatics intercomparison became a large portion of this work, and the authors note that the settings and nuances of the different informatics pipelines play a huge role in the metaproteomics results. I would therefore like to see the informatics methods for the labs come into the main text somehow (methods or a descriptive overview of the different informatics methods used), instead of only being in the supplement, even though they are long.

*Thank you for this suggestion. A new section 2.6 Bioinformatics Intercomparison Methods was added and the informatics methods were summarized. References from this section were added to the main document references as well.*

**Specific major comments**

Line 240: I'm hesitant on a blanket parent/fragment mass tolerance being applied the same way to all data, despite differences in mass analyzers and resolution for MS1 and MS2 used across the labs. Can the authors at least comment on this caveat?

*We incorrectly described our workflow, thank you for catching this. The corrected sentence is: "Specific parameters of the latter included: parent of tolerances of 10ppm were used on all instruments (all Orbitraps) for fragments tolerances of 0.02 Da or 0.6 Da were used for Orbitrap ms2 instruments and for ion trap ms2 instruments, respectively."*

Line 241/242: Percolator. What were the filtering settings used? Is it possible that there was a double FDR filtration (one in percolator and one in scaffold, which could result in fewer true positives being identified?)

*We checked on this topic and have added this sentence to the manuscript."Note that Scaffold ignores the Percolator output from Proteome Discoverer when re-running in Scaffold."*

Line 245: The protein inference and quantitation was conducted within Scaffold? Does this mean minimum evidence for a protein is 1 peptide? Was there any normalization applied?

*Yes 1 peptide per protein and no normalization was applied. No normalization was applied.*

Line 405: There is often discussion about whether perfectly matched metaG or metaT sequences are required for highest quality metaP analyses. Can the authors comment on why they believe that the same metagenome from the initial intercomparison study is still appropriate for the new samples taken a year later?

*The metaG sample for the laboratory study is a paired sample taken simultaneously with the proteomics sample. The reviewer is referring to the informatics intercomparison that uses the same database. We chose to use this database for uniformity within the study, but the reviewer is correct that an updated metaG sample might correspond better. However, the study region is known to have the same major taxon throughout the year allowing some consistenency between study components. The following text was added "This database was not collected simultaneously with the informatics samples, so it was not as representative as*

*that used in the laboratory intercomparison. However, the BATS study region is known to maintain similar major taxonomic composition throughout the year (e.g., Prochlorococcus and SAR11), hence enabling many protein identifications."*

Line 556: I think it's important to point out why ocean metaP will need to adopt metadata standards that are different than the ones that are already developed/being developed for other microbiome fields (i.e. the need to include geographical information, collection data such as filter sizes).

*Thank you for this suggestion.Our community did create metadata standards in a prior workshop and publication (Saito et al., 2021). A sentence was added to clarify and strengthen the data standardization needs. "While best practices for metadata and data types have been described by the community that include specific attributes important for environmental and ocean samples such as geospatial location and sample collection information (Saito et al., 2019) similar to the metadata standard recently put forward in the human proteome field (Dai et al., 2021), this study also demonstrated that there is a need for standardization of data output formats for metaproteomic results. "*

Data availability: Can the authors provide the annotation files for the metagenome fasta file? As currently provided in the pride submission, it would be impossible to re-analyze for function/taxonomy in the same way as the authors, and I could see that being of interest in the future i.e. for comparing annotation pipelines. This could be a flat file in the supplement or included in the genbank project.

*The annotation file is part of the PRIDE/ProteomeXchange submission. This is now explicitly described in the data availability section: "Data and Code Availability: The raw files, metagenome database (Intercal_ORFs_prodigal_metagenemark.fasta), and associated annotations (Intercal_assembly_annotations.csv) for this project summarized in Table S3 are available at ProteomeXchange and PRIDE repository with the dataset identifier PXD043218 and PXD044234."*

**Specific minor comments**

Line 74 – what is the measure of complexity being referenced? (dynamic range, proteins identified?)

*This was an introductory sentence describing the general application of proteomics to mixed environmental communities, rather than a statistical statement. We have clarified the sentence adding the parentheses in the following sentence: "Similar to other 'omics approaches, proteomics is increasingly being applied to natural ocean environments and the diverse microbial communities within them. When proteomics is applied to such mixed communities, it is generally referred to as metaproteomics (Wilmes and Bond, 2006)."*

Line 86 – Authors may find it useful to point out the particular usefulness of proteomics when applied to field/environmental samples, for revealing aspects of in situ biology that might not be resolved when organisms are isolated in the lab (e.g. Kleiner et al., 2019)

*We have added this topic and reference: "The functional perspective that metaproteomics allows is often complementary to metagenomic and metatranscriptomic analyses and can*

*provide biological insights that are distinct from organisms studied in the laboratory (Kleiner et al., 2019)."*

Line 123 – somewhere before or in this paragraph, it would be useful to further introduce that this study is focused on "discovery" or "global" or "shotgun" or "bottom up" metaproteomics, where the organisms and functions to be identified are not known in detail nor selected ahead of time

*This has been expanded on: "The effort focused on the data dependent analysis (DDA) methods, also known as global proteomics where the targets are unknown and hence there is a discovery element to the approach. DDA is currently common in ocean and other environmental and biomedical metaproteomics, and its associated spectral abundance units of relative quantitation have been shown to be reproducible in metaproteomics (Kleiner et al., 2017; Pietilä et al., 2022)."*

Line 173: Did lab 438 do the analysis or were the procedures of lab 438 performed in another lab?

*This has been clarified: "These samples were analyzed by 1D DDA analysis using extraction and mass spectrometry for laboratory 438 within their laboratory (Tables S5-S7)."*

Line 280 please add short detail about sample: e.g. North Atlantic Ocean 80m

*Additional information was added: "The laboratory study involved two biomass-laden filter slices collected from the North Atlantic Ocean Bermuda Atlantic Time series Study site at 80m depth being sent to each participating group for protein extraction, mass spectrometry, and informatic analyses (see Section 2.1). The informatic effort was independent of the laboratory effort and involved the distribution and informatic analysis of two metaproteomic raw data files generated from samples also from the North Atlantic Ocean upper water column BATS station (20m and 120m depths, see Section 2.1)."*

Line 290 Did the participants know anything at all about the samples, e.g. that they were from the oligotrophic open ocean?

*Yes the participants were provided with the sampling methods and metadata. This sentence has been added to the section: "Within both arms of the study, participants were provided the location of the study site and metadata about the sampling locations, time and depth at the onset of the study."*

Line 308: I'm interested in the proportion of shared peptides/proteins compared to the total number of unique peptides/proteins identified across the participants. Are identifications overlapping by 20%? 50%? In terms of the number of peptides/proteins ID'd when participants used their own pipelines, what was the range seen across laboratories?

*This information was added to the end of this sentence: "Despite these challenges, an average of 7142 +/- 2074 peptides were identified across the pairwise comparisons (Figure S1c) representing 20% of the 35,715 total unique peptides detected across all labs." The individual lab results are shown in Figure S1 as well. Peptide level results are shown, data output formats were so heterogenous as to make this comparison difficult on the protein level (e.g. protein groupings reported in some cases, or no protein inferred in other cases).*

Line 314: specific peptide and protein 1% FDR

*This sentence was added to provide these details "Raw data files were processed together within a single informatic pipeline consisting of SEQUEST-HT, Percolator, and Scaffold software and evaluated to a false discovery rate threshold of < 0.1% for peptides and 1.0% for proteins  (see Section 2.4)."*

Line 336: Does this imply that the number of peptides/proteins ID'd is a good measurement for overall data quality (especially if one wants to compare to other datasets)?

*Yes, we think this is a good metric for intercomparability. We have added a clause (in bold) to state this: "Laboratories with deeper proteomes shared numerous peptides, for example the two laboratories with the most discovered unique peptides shared ~3000 peptides between them, **implying that shared peptides is a useful metric for intercomparability**. They also had the largest numbers of peptides that were not found by any other labs (3617 and 2819, respectively)."*

Line 352: Related to above comment, among the deeper analyses, was the proportion of ID's that overlapped greater than the proportion in the 1D samples?

*We have not conducted shared peptide analyses on the deeper samples, using them only as an example of how the 1D approaches are constrained in depth. Given the length of the paper and how these deeper samples are not the focus, we are opting to avoid in depth analysis of these samples at this time and adding additional figures to the manuscript.*

Line 357: The sentence starting with "This indicated…" was difficult for me to understand. I suggest breaking into two sentences to clarify that if there was a fall off, it would be due to peptides being mapped to already discovered proteins.

*Thank you for this suggestion. One idea behind this analysis was a comparison to the rarefaction curve idea from DNA sequencing, where more sequencing did not result in saturation of depth. This sentence was clarified. "Together, the fact that there is a linear relationship between peptides and proteins across all laboratories (including labs employing deeper methods) could imply that the number of protein identifications has not begun to plateau and reached 'saturation', likely due to the immense biological diversity and abundance of lower abundance peptides within these samples. This approach has some similarities to rarefaction curves used in metagenomic sequencing to determine if the majority of species diversity has been sampled, although in this case number of peptides used as a metric for sampling depth instead of additional number of DNA sequencing samples typically used for rarefaction curves."*

Line 376: Is this quantitative consistency or the result of normalization?

*This consistency is not related to normalization, as none was applied to the dataset, including in the cross lab normalization. This is evident in the different slopes of the pairwise comparisons, which would be similar if they were normalized.*

*A clause at the end of this sentence was added to clarify this: "The slopes ranged from 0.33 to 5.5 (Figure S2), implying a varying dynamic range in spectral counts across laboratories,*

*likely due to variations in instrument parameterizations selected by each laboratory, and consistent with the lack of normalization between laboratories."*

Line 387: PstS from which organism?

*This sentence was edited (new text in bold): "Notably the PstS phosphate transporter protein **from Prochlorococcus** was the most abundant functional protein in all datasets, consistent with observations of phosphorus stress in the North Atlantic oligotrophic gyre and its biosynthesis in marine cyanobacteria (Scanlan et al., 1997; Coleman and Chisholm, 2010; Ustick et al., 2021)."*

Line 403: relative quantitative abundances

*We believe this comment is a suggested edit of line 430 rather than 403. We have replace the word 'quantitation' with the phrase 'quantitative abundances': "1) the identity of discovered peptides and proteins (Fig. 2), 2) their **relative quantitative abundances** (Figs. 4 and 6), and 3) the taxonomic and functional assignments within intercompared samples (Fig 5)."*

**Reviewer Comments 2**

Saito et al. present a comparison of metaproteomics sample preparation, measurement and analysis for ocean samples across several laboratories. The comparability of metaproteomics data across laboratories, especially of highly complex communities, is important to discuss, given also the increasing research efforts in this direction. As detailed in the manuscript, overall comparability is, despite a wide variety in applied methods, quite high between laboratories. At the same time, the manuscript also outlines room for further development.

General comments:

A clear rationale is given for focusing on 1D DDA, and for setting some boundaries for the analysis. The manuscript gives detailed insights into the comparisons. At the same time, it becomes at times unclear whether a reference is made to the informatics or wetlab comparison part of the study. Additionally, the manuscript would benefit from some shortening and focusing, especially in the discussion. For example, references to DIA are made at several places in the discussion, which could likely be condensed. On the other hand, some insights into reasons for differences would be interesting: Are there, e.g., indications that certain setups promote higher metaproteome coverage? Where there some re-runs by the same laboratory, which might be used to assess reproducibility within the same sample (given that it is highly complex) and laboratory?

*We appreciate these comments. New sentences were added to provide some lessons learned (see below). Re-runs were not conducted within this study.*

*Regarding the DIA text, two sentences were condensed into one: "Although the application of data independent approaches (DIA) to oceanographic metaproteomics analysis has been limited (e.g. Morris et al., 2010), the systematic nature of ion selection and fragmentation allows for a greater number of low abundant peptides to be quantified when enough ions can be isolated to produce robust MS2 spectra.,. "*

In addition, I second the comments of Reviewer 1 regarding the re-usability of data generated by different laboratories - which parameters could or should be fixed? These insights would also be immensely helpful in the strive for agreeing upon general (meta)proteomics standards. I would also be curious to see a brief comment of how one common search strategy might have impacted the results coming from different MS instruments with different settings. Also along the lines of comments of Reviewer 1, please increase the size of figures, figure captions and labels where appropriate.

*As mentioned above, new text was added regarding lessons learned from this study. We have word to increase the clarity of the figures.*

Specific comments:

Abstract

Line 43: I'd argue that metaproteomics has not only the potential for contribution to ocean ecology, but already did so - and would like to see some references for that, to make clear from the beginning why this manuscript is an important contribution to the field.

*Thank you for this comment. Line 43 is in the abstract, so not an appropriate place for a list of references in this journal's format. In the first paragraph of the introduction we do have a sentence with many references from metaproteomics studies included. The introductory paragraph has the following sentence, with minor edit in bold, including a list of metaproteomics references:* Proteomics (including metaproteomics) provides a perspective distinct from other 'omics methods: as a direct measurement of cellular functions it can be used to examine the diversity of ecosystem biogeochemical capabilities, to determine the extent of specific nutrient stressors by measurement of transporters or regulatory systems, to determine cellular resource allocation strategies in-situ, estimate biomass contributions from specific microbial groups, and even to estimate potential enzyme activity (Bender et al., 2018; Bergauer et al., 2018; Cohen et al., 2021; Fuchsman et al., 2019; Georges et al., 2014; Hawley et al., 2014; Held et al., 2021; Leary et al., 2014; McCain et al., 2022; Mikan et al., 2020; Moore et al., 2012; Morris et al., 2010; Saito et al., 2020; Sowell et al., 2009; Williams et al., 2012).

Line 52: While $R^2$=0.83 does indeed indicate good reproducibility, a value of 0.43 does less so - maybe give a (very brief) rationale for this discrepancy.

*Thank you for this comment. Based on this suggestion we explored the data again. It looks like one laboratory using an older instrument and discovering fewer peptides (209) when compared to two other laboratories with long dynamic exclusion times resulted in the lowest $R^2$ values. The longer dynamic exclusion times favor increased peptide discovery, but at the expense of acquiring counts for the more abundant peptides, and hence these pairwise comparisions resulted in lower correlation coefficients.*

*We also revisited the range of $R^2$ values. We calculated the average and standard deviation of correlation coefficients. This value of 0.63 +/- 0.11 was more representative of the groups' results and replaced the range of values in the abstract. We also added a new supplemental table S12 to summarize all of these values that was for some new calculations of results:*

**Table S12. Summary Table of Laboratory Intercomparison Results**

| | Lab_127 | Lab_135 | Lab_209 | Lab_438 | Lab_652 | Lab_729 | Lab_774 | Average | Std Dev |
|---|---|---|---|---|---|---|---|---|---|
| Sum of Spectral Counts | 73828 | 38784 | 63198 | 126642 | 69677 | 53166 | 70606 | 70843 | 27455 |
| Number of Peptide IDs | 12615 | 9600 | 3354 | 15646 | 9106 | 6626 | 16500 | 10492 | 4757 |
| Number of Protein IDs | 5080 | 3919 | 1586 | 6221 | 3518 | 3522 | 5676 | 4217 | 1574 |
| Number of Protein Groups | 4595 | 3533 | 1461 | 5621 | 3189 | 3202 | 5111 | 3816 | 1411 |
| | | | | | | | | Average* | Std Dev* |
| Average Shared Peptides (pairwise 7 labs) | 2821.0 | 2422.8 | 1304.2 | 2945.0 | 2325.7 | 2241.5 | 2769.2 | 2404 | 554 |
| Average R2 (pairwise 7 labs)* | 0.708 | 0.586 | 0.589 | 0.713 | 0.652 | 0.604 | 0.583 | 0.63 | 0.06 |
| Average Slope (pairwise 7 labs) | 1.099 | 3.014 | 1.617 | 1.386 | 1.297 | 1.028 | 0.710 | 1.45 | 0.75 |
| Dynamic Exclusion Time (s) | 30 | 60 | 30 | 15 | 10 | 30 | 60 | 33.57 | 19.73 |

*average and standard deviation of all pairwise comparisons

*To describe these additional analyses we have added the following text. As this analysis contributes to the lessons learned requested by reviewers, we felt this was a reasonable cause for additional text:*

*"While there is good agreement, this congregation of data allows further exploration of the influence of methods on the results. In particular, as mentioned above the range of pairwise comparisons had correlation coefficients ranging from 0.43 to 0.84, with most values falling between 0.6 and 0.8 (Figure 4b and 4e; Table S12). This average of all correlation coefficients described above (0.63 +/- 0.11) implied good reproducibility between laboratories in general. We can explore what might have influenced the variability and lower range of coefficients. The correlation coefficients of lab 209 had two of the three $R^2$ values below 0.499 in pairwise comparisons (0.431 and 0.475), yet also had values that ranged from 0.61 to 0.70. Why would this variability exist? Lab 209 's methods differed from other labs in several ways: they used the oldest and slowest instrument of the group (Thermo Orbitrap Elite), used CID instead of HCD for fragmentation and rapid scan mode, and used an unusually long column of 200cm to compensate for the older instrument (Table S6). As a result, lab 209 had the lowest number of peptide (3354) and protein (1586) ID's of the seven labs (Table S9), which was several fold lower than the lab with the highest number and reduced the number of shared peptides across all laboratories. In pairwise comparisons, lab 209 had the lowest number of shared peptides at an average of 1304. Interestingly however, lab 209 did not have the lowest number of total spectral counts (63198), being close to the average (70843 +/- 27455), implying that more abundant peptides were detected relative to rarer ones.*

*We initially suspected the lower $R^2$ values in pairwise comparisons with lab 209 may have been related to comparisons to laboratories with similarly lesser peptide depth, but this was not the case: the two lowest correlation coefficients for lab 209 were with laboratories 135 and 774 (the 0.431 and 0.475 values), the latter of which had the highest number of peptide identifications. The answer for this difference in quantitative values maybe within the selection of parameters used to sample peptide peaks: Both lab 135 and 774 used 60 second dynamic exclusion, whereas the other 5 labs used dynamic exclusions between 10 and 30 seconds in length (Table S7). This higher dynamic exclusion likely contributed to providing greater peptide discovery depth, but at the cost of quantitative consistency with other laboratories, since this parameter selects against repeat counting of abundant peaks and would reduce spectral counts of the more abundant peptides that lab 209 was detecting. This result demonstrates the influence of the mass spectrometer parameters in quantitative reproducibility when using global proteomic DDA mode. "*

The term "informatic" appears a bit ambiguous at times - maybe consider replacing this by "bioinformatic" or "computational".

*The term informatic was changed to bioinformatic throughout the manuscript, except when referring to the Proteomics Informatics Group (iPRG).*

Line 58: To what does reproducibility refer here?

*The idea is that this kind of study could be expanded from our constrained study (1D analyses) to unconstrained options that included deeper analyses.*

Introduction

Lines 66-67: please give references.

*References added: (Falkowski et al., 2008; Moran et al., 2022; Worden et al., 2015).*

Lines 73 ff: There is a jump in content: first, metaproteomics is introduced, and then proteomics (not metaproteomics) is detailed again.

*This was clarified to include metaproteomics: "Proteomics (including metaproteomics) provides a perspective distinct from other 'omics methods"*

Lines 83-86: Maybe check whether a focus can be set specifically for marine environments

*The sentence as written doesn't state proteomics and metaproteomics applications are limited to marine examples, we prefer to leave this as is with the minor edit in bold. "Proteomics **(including metaproteomics)** provides a perspective distinct from other 'omics methods: as a direct measurement of cellular functions it can be used to examine the diversity of ecosystem biogeochemical capabilities, to determine the extent of specific nutrient stressors by measurement of transporters or regulatory systems, to determine cellular resource allocation strategies in-situ, estimate biomass contributions from specific microbial groups, and even to estimate potential enzyme activity"*

Line 88: "measurement of microbial proteins" - this could refer to many different methods, but probably refers to metaproteomics? Please formulate more specifically.

*This sentence has been edited with the added text in bold: "Moreover, the measurement of microbial proteins **in environmental samples** has improved greatly in recent years, due to the advancements in nanospray-liquid chromatography and high-resolution mass spectrometry approaches (Mueller and Pan, 2013; Ram et al., 2005; McIlvin and Saito, 2021)."*

Line 92: please consider replacing "the metaproteomics datatzpe", e.g., by "metaproteomics data"

*Done.*

Methods

Line 245: There seems to be a word missing after "protein"

*Thank you for catching that. This has been edited removing "The protein in this" clause and replacing it with the bold text: "**The re-analysis was** conducted within Scaffold using total spectral counts and allowing single peptides to be attributed to proteins."*

Results

Line 273, 277: please replace "activities" and "aims" with components (or another consistently used term) to aid in understanding. Potentially, you could add separate names or labels for the two study part.

*Done – components replaced activities throughout.*

Lines 314-315: please remove the surplus "see"

*Done.*

Line 340: Do you mean "shared" instead of "showed"?

*Yes, corrected, thank you.*

Lines 342-346: This sentence is somewhat hard to read, maybe split.

*This sentence was edited for clarity to : "A similar analysis was conducted at the protein level, where the number of proteins identified (see Section 2. Methods) identified 8,043 unique proteins in total across all laboratories, with 1,056 proteins of those observed in all seven labs (see 7-way Venn diagram in Figure 2c). Three-way Venn diagram comparisons among labs 135, 209 and 438 had an intersection of 1,254 proteins, and labs 652, 729, and 774 shared 1,925 proteins (data not shown)."*

Line 364: Do you mean "consensus" in place of "coherence"?

*We intended coherence, to imply the consistency in shared proteins.*

Lines 367-368: Isn't a unity line always observed when comparing a dataset to itself?

*Yes, we just stated it to avoid any confusion. This was mildly edited (as expected). "When a dataset was compared with itself a unity line of datapoints was observed along the diagonal axis as expected"*

Line 374: Do you mean "consensus" in place of "coherence"?

*We intended coherence, to imply the consistency in shared proteins.*

Line 383: How abundant are these organisms? Please give some estimations.

*Done. The following text was added: "(Sowell et al., 2009; Malmstrom et al., 2010). For example, Prochlorococcus is consistently present between $10^4$ and $10^5$ cells per milliliter in this region and has been observed to contribute to carbon export from the euphotic zone (Casey et al., 2007). Pelagibacter cells can also be in excess of $10^5$ cells per milliliter at the BATS North Atlantic location (Carlson et al., 2009)."*

*With the following references added to the manuscript:*

*Carlson, C.A., Morris, R., Parsons, R., Treusch, A.H., Giovannoni, S.J. and Vergin, K., 2009. Seasonal dynamics of SAR11 populations in the euphotic and mesopelagic zones of the northwestern Sargasso Sea. The ISME journal, 3(3), pp.283-295.*

*Casey, J.R., Lomas, M.W., Mandecki, J. and Walker, D.E., 2007. Prochlorococcus contributes to new production in the Sargasso Sea deep chlorophyll maximum. Geophysical Research Letters, 34(10).*

Line 387: do you mean proteins or protein functional groups, as this refers to a KEGG analysis?

*Added the word "functional" to : "Similarly, KEGG Orthology group (KO) analysis of those datasets also showed highly similar patterns of protein functional distributions across laboratories (Figure 5b)." Thank you for the suggested clarification.*

Line 388: What does "functional" refer to here?

*Deleted the word functional here (befween words* abundant protein*). "Notably the PstS phosphate transporter protein from Prochlorococcus was the most abundant protein in all datasets, consistent with observations of phosphorus stress in the North Atlantic oligotrophic gyre and its biosynthesis in marine cyanobacteria"*

Lines 394 ff: Please re-phrase/simplify, e.g., Variability at the protein level was lower than at the peptide level

*This has been rewritten as: "When conducted on the full dataset with all peptides and proteins, the Sørensen similarity analyses showed peptides had lower similarity than proteins, implying variability is ameliorated when aggregated to the protein level (Figure S3)."*

Lines 404-405: Please briefly elaborate on the use of the same database, given that the North Atlantic is a highly variable and shifting environment

*Additional text was added: "This database was not collected simultaneously with the bioinformatics samples, so it was not as representative as that used in the laboratory intercomparison. However, the BATS study region is known to maintain similar major taxonomic composition throughout the year (e.g., Prochlorococcus and SAR11), hence enabling many protein identifications."*

Lines 410 ff: Maybe the general pipelines could be briefly presented in the main text.

*We have added more bioinformatics methods to the main text, and is referred to in this section:* "The bioinformatic intercomparison involved 10 laboratories utilizing 8 different software pipelines including the PSM search engines: SEQUEST, X!Tandem, MaxQuant, MSGF+, Mascot, MSFragger, and PEAKS (Table S11**, see Methods Section 2.6**)."

Discussion

Lines 424-426: Please shorten and focus this sentence.

*The sentence was edited for clarity, thank you. "Given the recent establishment of complex metaproteomic techniques, intercomparisons are valuable in demonstrating their suitability for ocean ecological and biogeochemistry studies."*

Lines 434-437: Please shorten.

*The sentence was shortened to: "These results provide confidence that multiple laboratories can generate reproducible results describing the major proteome composition of ocean microbiome samples to assess their functional and biogeochemical activity"*

Line 439: Did you use a specific cutoff to determine abundant proteins?

*This is evident in Figure 8, which was introduced at the end of the paragraph. This sentence has been moved to the second sentence to support the introduction sentence about abundant proteins being shared between the labs. "This is evident in Figure 8, where most of the 1063 proteins across seven laboratories in the re-analysis were in the upper half of proteins when ranked by abundance."*

Lines 440 ff. Please formulate more clearly, e.g., "Probable reasons for this discrepancy are: …"

*This sentence was shortened, but the distinction about abundant and rare proteins was kept: "This simultaneous consistency in abundant proteins and diversity in rare proteins (and their respective peptide constituents) was likely a result of several factors."*

Lines 478-485: Please shorten.

*We respectfully request to keep this text. A complex idea is being articulated here that is specific to the challenges of metaproteomics. These five sentences briefly outline this issue.*

**Citation**: https://doi.org/10.5194/egusphere-2023-3148-RC2

---

## Author Response (AR2)

The following two figures have been improved for clarity. Thank you.

Figure 4.

[Figure]

[Figure]

**Figure 5.**

[Figure]